# FEDFUSE: SELECTIVE KNOWLEDGE DISTILLATION WITH EXPERT-GUIDED FUSION FOR HETEROGENEOUS FEDERATED LEARNING

## ABSTRACT

Heterogeneous Federated Learning enables collaborative training across devices with diverse architectures and non-IID data. However, it struggles with effective knowledge fusion, leading to personalized knowledge loss during aggregation and client model divergence due to globally-guided updates misaligned with local data or architectures. We propose FedFuse, a novel framework for adaptive, personalized knowledge fusion via logits. FedFuse introduces a server-side Expert-guided Fusion mechanism that facilitates adaptive knowledge fusion by dynamically gating and weighting heterogeneous client knowledge contributions, moving beyond static schemes. Complementarily, a selective knowledge distillation strategy allows clients to assimilate global knowledge without blind imitation, preserving crucial local features and mitigating model divergence. We provide rigorous convergence analysis for FedFuse under heterogeneity. Extensive experiments, including up to 500 clients, diverse heterogeneity settings, and ablation studies, demonstrate our approach's superiority. FedFuse significantly outperforms state-of-the-art methods in test accuracy, particularly under high heterogeneity, while maintaining competitive efficiency.

## 1 INTRODUCTION

Heterogeneous Federated Learning (HeteroFL) presents a compelling paradigm for collaborative machine learning in diverse edge computing environments, such as the Artificial Intelligence of Things Zhang et al. (2020), smart surveillance Pang et al. (2023), autonomous vehicles Nguyen et al. (2022). It uniquely accommodates the reality of edge ecosystems where clients possess varying computational resources, data distributions (statistical heterogeneity), and even distinct underlying model architectures (architectural heterogeneity) tailored to local needs. While HeteroFL effectively leverages the collective knowledge of the heterogeneous network, it introduces a critical issue: the loss of personalized knowledge. This phenomenon, where discrepancies arise between local objectives and global aggregation due to client heterogeneity significantly degrades the performance of personalized models.

While many approaches offer valuable contributions, two key challenges remain. First, the primary challenge is how to effectively fuse heterogeneous client knowledge while preserving personalized features during aggregation. Conventional methods Sattler et al. (2021); Zhu et al. (2021b); Zhang et al. (2024); Jang et al. (2022) typically produce a single homogenized global representation, which makes it difficult to differentiate and preserve personalized knowledge, particularly for clients with divergent data distributions. Consequently, valuable knowledge is often diluted or ignored, degrading the performance of the fused global model. The second challenge is how to align global knowledge with the local model without disrupting locally learned features, which are essential for maintaining personalized performance. Traditional methods enforce uniform updates that may conflict with client heterogeneity, forcibly diverting the local model from its optimized state.

To tackle these intertwined challenges of adaptive aggregation and compatible personalization, we propose FedFuse, a novel HeteroFL framework built upon two synergistic insights: leveraging expert-guided fusion and utilizing selective knowledge distillation. The choice of an expert-guided fusion mechanism for server-side knowledge fusion is motivated by its potential to handle heterogeneity

effectively, drawing inspiration from its success in large-scale modeling Shen et al. (2024); Shazeer et al. (2017). We introduce a server-side MoE operating on uploaded logits Shazeer et al. (2017). Unlike static aggregation, the MoE's dynamic gating network learns to route and weight knowledge contributions (represented by logits) from different clients to specialized experts. This allows the server to adaptively capture relevant knowledge patterns from subsets of clients, preserving personalized information while constructing a rich, diverse global representation.

Furthermore, FedFuse incorporates a selective knowledge distillation strategy designed for compatibility with local models. Instead of blindly applying global updates, clients selectively integrate only the most relevant global knowledge, determined by the alignment between local and global feature representations, minimizing negative transfer and preserving crucial local features. This selective approach operates within the logits space Gou et al. (2021), further ensuring compatibility between local and global updates and mitigating the risk of disrupting locally learned features.

The **main contributions** of this paper are therefore summarized as:

- We propose FedFuse, a framework enabling expert-guided fusion and selective knowledge distillation across statistically and architecturally heterogeneous clients.
- We introduce a novel expert-guided fusion mechanism that dynamically captures and fuses personalized knowledge from heterogeneous clients based on relevance between local knowledge and global experts, mitigating personalization loss during aggregation.
- We introduce a selective knowledge distillation strategy that selects favorable global knowledge for local model updates to preserve key local model features.
- We conduct extensive empirical validation across diverse benchmarks (CIFAR-100, Tiny-ImageNet, Flower102), including large-scale scenarios with up to 500 clients and rigorous ablation studies, demonstrating significant accuracy improvements over state-of-the-art HeteroFL methods, particularly under high heterogeneity, while maintaining competitive resource efficiency.

## 2 RELATED WORK

Federated learning under statistical and architectural heterogeneity (HeteroFL) has garnered significant attention. Existing approaches primarily fall into three categories.

**HeteroFL with Knowledge Distillation (KD).** These methods Sattler et al. (2021); Zhu et al. (2021a); Song et al. (2024); Yao et al. (2023); Gong et al. (2024); Li & Wang (2019); Ma et al. (2022); Lin et al. (2020) leverage knowledge distillation, where clients typically train local models and generate knowledge representations (e.g., soft labels, feature maps) from their private data Jeong et al. (2018). These representations are aggregated by the server to guide the training of client models (students) or a global model, avoiding direct parameter sharing. Examples include FedGKD Yao et al. (2023) and FedIOD Gong et al. (2024). While effective for basic knowledge fusion, a key limitation arises in personalization: the aggregation process often distills knowledge into a single, potentially homogenized teacher model. This averaged knowledge may struggle to adequately capture or fuse the specialized, personalized features required by clients with highly diverse data distributions or functional roles within the HeteroFL network.

**HeteroFL with Lightweight Representations.** To reduce communication overhead and handle architectural diversity, some methods employ lightweight representations instead of full model parameters for aggregation. One line of work uses prototypes Dai et al. (2023); Tan et al. (2022); Zhang et al. (2024), where clients upload class prototypes derived from their local data, which are then aggregated by the server. Another approach involves sharing intermediate feature representations Huang et al. (2022); Yi et al. (2023; 2024b), allowing clients to contribute learned features rather than parameters. While these methods significantly reduce communication costs, relying on such simplified or aggregated representations (prototypes or features) carries the risk of information bottleneck, potentially losing the fine-grained details crucial for deep personalization on individual clients. It remains challenging for these compact representations to fully encapsulate the diverse functionalities and specificities present across a truly heterogeneous client network.

**HeteroFL with Model Transformation.** This category focuses on aligning heterogeneous model structures for aggregation. Some methods split models into shared components (e.g., feature extrac-

tors) and personalized components (e.g., predictors) Collins et al. (2021); Oh et al. (2021); Chen et al. (2021); Pillutla et al. (2022); Liu et al. (2022); Jang et al. (2022); Liang et al. (2020). Others attempt to standardize heterogeneous architectures into a common format before aggregation or matching Diao et al. (2021); Wang et al. (2020; 2024). While enabling collaboration across different architectures, model transformation often imposes structural constraints, such as requiring a uniform feature extractor dimension or specific layer types. This can limit the flexibility needed for clients with genuinely distinct hardware capabilities or highly specialized local tasks, potentially hindering optimal local adaptation and personalization. The transformation or matching process itself might also inadvertently discard valuable model-specific information pertinent to a client's unique role.

In a nutshell, while prior research has made significant strides, existing paradigms often face difficulties in effectively balancing global knowledge fusion with local personalization. Methods based on KD can risk generating overly generalized guidance, lightweight representations may lack sufficient granularity for deep personalization, and model transformations can impose restrictive structural constraints. These limitations hinder the ability to *adaptively aggregate* diverse personalized knowledge without significant information loss and to subsequently *compatibly disseminate* relevant global insights without disrupting local model specialization. These challenges collectively contribute to the persistent problem of client heterogeneity. To overcome these specific shortcomings, FedFuse introduces a different approach. Our expert-guided fusion mechanism directly tackles the adaptive fusion challenge by dynamically identifying and weighting relevant client knowledge via expert gates, moving beyond simplistic averaging or static representations. Moreover, the compatible personalization challenge is addressed by our selective knowledge distillation strategy, specifically designed to integrate this tailored global knowledge while respecting local model integrity and specificity.

## 3 METHODOLOGY

### 3.1 PROBLEM FORMULATION

Consider a federated learning setting with $K$ clients, indexed by $i = 1, \ldots, K$. Each client $i$ possesses a private dataset $\mathcal{D}_i^r$, which typically exhibits non-IID characteristics across clients, leading to statistical heterogeneity. Furthermore, each client maintains a local model $M_i$, parameterized by $\theta_i$. These models $M_i$ can vary significantly in terms of architecture, depth, or capacity (architectural heterogeneity), reflecting diverse device capabilities and local requirements. The overarching goal is collaborative training to enhance each client's personalized model performance on its own data, rather than converging to a single global model.

Formally, the ideal personalized objective can be conceptualized as minimizing a collective loss function over the private datasets:

$$\min_{\{\theta_i\}_{i=1}^K} \sum_{i=1}^K p_i \mathcal{F}_i(\theta_i), \quad \text{where} \quad \mathcal{F}_i(\theta_i) = \mathbb{E}_{(x,y) \sim \mathcal{D}_i^r}[\mathcal{L}(M_i(\theta_i; x), y)] \tag{1}$$

Here, $\mathcal{L}$ is a loss function (e.g., cross-entropy), $M_i(\theta_i; x)$ is the prediction of client $i$'s model, and $p_i$ is a weighting factor (e.g., proportional to $|\mathcal{D}_i^r|$ or $1/K$).

However, FedFuse does not directly optimize this ideal objective due to the challenges of heterogeneous knowledge fusion. Instead, our framework employs a three-stage approach with distinct loss functions for different training phases:

$$\text{Local Training: } \mathcal{L}_c, \quad \text{Server Fusion: } \mathcal{L}_a, \quad \text{Personalization: } \mathcal{L}_p \tag{2}$$

The constituent loss terms $\mathcal{L}_c$, $\mathcal{L}_a$, and $\mathcal{L}_p$ correspond to client local training, server-side expert-guided fusion, and client-side selective knowledge distillation, respectively. These will be elaborated in Sections 3.3, 3.4, and 3.5.

A core challenge in HeteroFL is how to effectively fuse heterogeneous client knowledge preserving personalized features during aggregation. The averaged fusion of global knowledge frequently leads to dilution or loss of valuable information, consequently degrading the performance of integrated global knowledge.

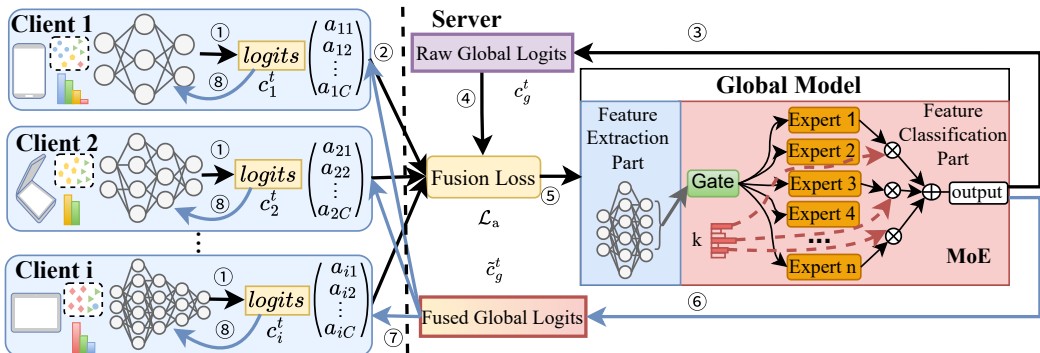

Figure 1: Overview of FedFuse architecture. ①,② Clients generate logits $c_i^t$ using public dataset and upload them. ③ Server's global MoE model produces logits $c_g^t$. ④, ⑤ Fusion loss $\mathcal{L}_a$ updates MoE parameters using client and global logits. ⑥ Updated MoE generates refined global logits $\tilde{c}_g^t$. ⑦, ⑧ Clients use $\tilde{c}_g^t$ for personalized updates via selective knowledge distillation.

Access to a small, publicly available dataset $\mathcal{D}^b$ is assumed to facilitate model-agnostic knowledge fusion (e.g., via logits) without compromising data privacy. This public dataset serves as a common reference for knowledge representation and is a key enabler for our proposed mechanisms. While this assumption may limit applicability in scenarios where no suitable public data exists, it is commonly adopted in federated learning literature and reflects practical scenarios where public datasets (e.g., ImageNet for vision tasks) are available for the target domain.

## 3.2 THE OVERVIEW OF FEDFUSE FRAMEWORK

To address adaptive aggregation and compatible personalization challenges in HeteroFL, we propose FedFuse. The overall architecture, illustrated in Figure 1, orchestrates cyclical knowledge flow: from clients to server for fusion, then back to clients for personalized guidance. This framework comprises three main stages per communication round: (1) **Client Local Training and Knowledge Representation:** Clients perform local training on private data $\mathcal{D}_i^r$, then compute output logits on public dataset $\mathcal{D}^b$. These model-agnostic logits, capturing current knowledge state, are sent to the server. (2) **Expert-guided Fusion:** The server employs a Mixture-of-Experts (MoE) mechanism on received client logits. A gating network dynamically selects and weights relevant experts to process features from $\mathcal{D}^b$, producing aggregated global logits that adaptively fuse diverse knowledge from heterogeneous clients. (3) **Selective Knowledge Distillation:** The server generates refined global logits and distributes them to clients. Each client uses these global logits to guide local model parameter updates via reverse KL divergence, integrating global insights while preserving local model specificity. This design achieves effective knowledge sharing tailored to diverse client needs, mitigating model divergence and enhancing personalized performance.

## 3.3 CLIENT LOCAL TRAINING AND KNOWLEDGE REPRESENTATION

In communication round $t$, each participating client $i$ (from a selected subset $N_t \subseteq \{1, \ldots, K\}$) first updates its local model parameters $\theta_i^t$ using its private data $\mathcal{D}_i^r$. This local training typically involves multiple steps of gradient descent:

$$\theta_i^t \leftarrow \theta_i^{t-1} - \eta \nabla_{\theta_i^{t-1}} \mathbb{E}_{(x,y) \sim \mathcal{D}_i^r} \left[ \mathcal{L}_c(f_{\theta_i^{t-1}}(x), y) \right], \tag{3}$$

where $f_{\theta_i}$ is the forward pass of model $M_i$, $\mathcal{L}_c$ is the cross-entropy loss, and $\eta$ is the local learning rate.

Subsequently, client $i$ uses $f_{\theta_i^t}$ to compute output logits on the public dataset $\mathcal{D}^b$. The raw output logits $c_{i,l}^t$ from client $i$ for input $x$ at round $t$:

$$c_{i,l}^t = f_{\theta_i^t} \in \mathbb{R}^C, \tag{4}$$

where $C$ is the number of classes in $\mathcal{D}^b$. We apply temperature scaling for better knowledge fusion:

$$c_i^t = \mathrm{softmax}\left(\frac{c_{i,l}^t(x)}{\tau}\right), \tag{5}$$

where $\tau$ is the temperature parameter ($\tau = 2.0$ in our experiments). The set of logits $\{c_i^t | x \in \mathcal{D}^b\}$ for client $i$ is uploaded to the server.

### 3.4 EXPERT-GUIDED FUSION

Upon receiving logits $\{c_i^t\}_{i \in N_t}$ from participating clients, the server employs an MoE mechanism to adaptively fuse this knowledge. The server maintains a global model $M_g$ to learn the fusion function, with the first fully connected layer(s) structured as the MoE layer.

Let $\theta_g^t$ denote the parameters of the global model at round $t$. For an input $x \in \mathcal{D}^b$, let $\psi_{\theta_g^t}(x) \in \mathbb{R}^{d_{in}}$ be the output of the feature extractor $\psi_{\theta_g^t}$ of $M_g$. $\psi_{\theta_g^t}(x)$ is fed into the MoE layer, which consists of $E$ parallel experts ($e_j : \mathbb{R}^{d_{in}} \to \mathbb{R}^C$) and a gating network $g(\cdot) : \mathbb{R}^{d_{in}} \to \mathbb{R}^E$. The gating network computes scores for each expert based on the input features:

$$g(\psi_{\theta_g^t}(x)) = W_g \psi_{\theta_g^t}(x) + b_g, \tag{6}$$

where $W_g \in \mathbb{R}^{E \times d_{in}}$ and $b_g \in \mathbb{R}^E$ are parameters of the gating network. To achieve sparse activation, which is often preferred in MoE for efficiency and specialization, we employ a top-$k$ gating strategy(top-$k$ sensitivity experiments in the Appendix E). The gate selects the set $\mathcal{E}(x)$ containing the indices of the $k$ experts with the highest scores in $g(\psi_{\theta_g^t}(x))$, where $k \ll E$. The routing weights $\pi_j$ are then computed via softmax over the scores of the selected experts:

$$\pi_j(\psi_{\theta_g^t}(x)) = \begin{cases} \frac{\exp(g_j(\psi_{\theta_g^t}(x)))}{\sum_{l \in \mathcal{E}(x)} \exp(g_l(\psi_{\theta_g^t}(x)))}, & \text{if } j \in \mathcal{E}(x), \\ 0, & \text{otherwise.} \end{cases} \tag{7}$$

This dynamic, input-dependent selection allows the model to route different inputs from $\mathcal{D}^b$ to potentially different subsets of experts, enabling specialized knowledge processing. Each selected expert $j \in \mathcal{E}(x)$ processes the input features:

$$e_j(\psi_{\theta_g^t}(x)) = W_j \psi_{\theta_g^t}(x) + b_j, \tag{8}$$

where $W_j \in \mathbb{R}^{C \times d_{in}}, b_j \in \mathbb{R}^C$ are the parameters of expert $j$. The final MoE output (pre-activation global logits) is a weighted combination of the outputs from the selected experts:

$$\mathrm{MoE}(\psi_{\theta_g^t}(x)) = \sum_{j \in \mathcal{E}(x)} \pi_j(\psi_{\theta_g^t}(x)) e_j(\psi_{\theta_g^t}(x)). \tag{9}$$

Similar to the client-side processing, we apply temperature scaling to get the global logits:

$$c_g^t = \mathrm{softmax}\left(\frac{\mathrm{MoE}(\psi_{\theta_g^t}(x))}{\tau}\right). \tag{10}$$

The core idea is to train the global MoE model ($\theta_g$) such that its output distribution $c_g^t$ optimally reflects a fusion of the class distributions $\{c_i^t\}_{i \in N_t}$. We achieve this by minimizing the average Kullback-Leibler (KL) divergence Kullback & Leibler (1951) from the client logits to the global logits over the public dataset and participating clients. The choice of $D_K(P \parallel Q)$ aims to find a $Q$ (global logits) that is close to the average of $P$s (client logits) in terms of information content. The fusion loss is:

$$\mathcal{L}_a = \frac{1}{|N_t||\mathcal{D}^b|} \sum_{i \in N_t} \sum_{x \in \mathcal{D}^b} \tau^2 D_K(c_i^t \parallel c_g^t). \tag{11}$$

The KL divergence Hinton et al. (2015) is calculated as:

$$D_K(P \parallel Q) = \sum_{c=1}^{C} P_c \log \frac{P_c}{Q_c}. \tag{12}$$

---

**Algorithm 1** Expert-Guided Fusion

---

**Require:** Current global model $\theta_g^{t-1}$, local client logits $\{c_i^t\}_{i \in N_t}$, public dataset $\mathcal{D}^b$, server epochs $E_s$, global learning rate $\eta_g$.
1: **Expert-Guided Fusion:**
2: Let $\theta_g^{t-1}$ represents the global model parameters at round $t-1$.
3: **for** server epoch $e = 1, \ldots, E_s$ **do**
4:    Compute global logits $c_g^t = \{c_g^t(x) | x \in \mathcal{D}^b\}$ via Equation 10 using $\theta_g^{t-1}$
5:    Calculate fusion loss $\mathcal{L}_a$ via Equation 11 using $\{c_i^t\}_{i \in N_t}$ and $c_g^t$
6:    Update $\theta_g^{t-1}$ using Equation 13: $\theta_g^t \leftarrow \theta_g^{t-1} - \eta_g \nabla_{\theta_g^{t-1}} \mathcal{L}_a$
7: **end for**
8: Let $\theta_g^{t-1} \leftarrow \theta_g^t$
9: Compute updated global logits $\tilde{c}_g^t = \{\tilde{c}_g^t | x \in \mathcal{D}^b\}$ via Equation 14 using $\theta_g^t$
10: **return** $\tilde{c}_g^t, \theta_g^t$

---

The server updates the global model parameters $\theta_g$ using gradients from this aggregation loss:

$$\theta_g^t \leftarrow \theta_g^{t-1} - \eta_g \nabla_{\theta_g^{t-1}} \mathcal{L}_a, \tag{13}$$

where $\eta_g$ is the server learning rate. Note that the server only needs the client logits $c_i^t$, not the client model parameters $\theta_i^t$. This part is shown in Alogorithm 1.

### 3.5 SELECTIVE KNOWLEDGE DISTILLATION

After updating the global MoE model to $\theta_g^t$, the server uses it to generate a refined set of global logits $\tilde{c}_g^t$ for distribution back to the clients. These are computed using the updated parameters $\theta_g^t$ on $\mathcal{D}^b$:

$$\tilde{c}_g^t = \text{softmax}\left(\frac{\text{MoE}(\psi_{\theta_g^t}(x))}{\tau}\right). \tag{14}$$

Each client $i$ receives this set of global logits $\tilde{c}_g^t = \{\tilde{c}_g^t | x \in \mathcal{D}^b\}$. The client then performs a local update step aimed at incorporating the global knowledge encoded in $\tilde{c}_g^t$ while retaining its personalized features. This is achieved by minimizing a loss function that encourages the client's logits $c_i^t$ (computed using its current parameters $\theta_i^t$) to align with the received global logits $\tilde{c}_g^t$. We employ the reverse KL divergence for this purpose:

$$\mathcal{L}_p = \frac{1}{|\mathcal{D}^b|} \sum_{x \in \mathcal{D}^b} D_K(\tilde{c}_g^t \parallel c_i^t). \tag{15}$$

The choice of reverse KL divergence $D_K(Q \parallel P)$ is intentional and crucial for personalization. Minimizing $D_K(Q \parallel P)$ encourages $P$ (client logits) to have high probability where $Q$ (global logits) has high probability, but allows $P$ to maintain its own modes (preserving personalization) where $Q$ has low probability. This contrasts with minimizing the forward KL divergence $D_K(P \parallel Q)$, which tends to force $P$ to cover all modes of $Q$, potentially suppressing $P$'s unique features that are essential for local personalization. This update step modifies the client parameters:

$$\theta_i^t \leftarrow \theta_i^{t-1} - \eta_p \nabla_{\theta_i^{t-1}} \mathcal{L}_p, \tag{16}$$

where $\eta_p$ is the learning rate for the personalization update. $\theta_i^t$ becomes the starting point for the next communication round's local training on $\mathcal{D}_i^r$. The complete process is summarized in Algorithm 2.

### 3.6 THEORETICAL ANALYSIS

In this subsection, we provide theoretical insights into the proposed FedFuse framework. We establish convergence guarantees for the algorithm under standard assumptions commonly used in federated optimization, demonstrating its stability and convergence properties in the HeteroFL setting. Our analysis considers both strongly convex and non-convex cases, accounting for the composite loss structure involving client training ($\mathcal{L}_c$), server-side MoE fusion ($\mathcal{L}_a$), and personalization updates ($\mathcal{L}_p$). The detailed theorems, assumptions, and proofs for the convergence analysis are presented in Appendix C.2.

**Theorem 1** (Convergence for Strongly Convex Case)*. Under Assumptions 1, 2, 3, and 4, with appropriate learning rates, FedFuse satisfies:*

$$E[F(\theta^t) - F(\theta^*)] \leq \frac{1}{2\mu\eta T}[F(\theta^1) - F(\theta^*)] + \frac{\eta\sigma^2}{4\mu}(L + \beta + \gamma),\tag{17}$$

*where $L$, $\beta$, $\gamma$ are smoothness parameters for $\mathcal{L}_c$, $\mathcal{L}_a$, $\mathcal{L}_p$ respectively, $\mu$ is the strong convexity parameter, and $\sigma^2$ bounds the stochastic gradient variance.*

**Theorem 2** (Convergence for Non-Convex Case)*. Under Assumptions 1 and 3, FedFuse satisfies:*

$$E[\|\nabla F(\theta^t)\|^2] \leq \frac{2}{\eta T(1 - \eta L)}(F(\theta^1) - F(\theta^*)) + \frac{3\eta L\sigma^2}{2 - \eta L}.\tag{18}$$

## 4 EXPERIMENTS

**Datasets.** We evaluate on three CV datasets: CIFAR-100 Krizhevsky (2009), TinyImageNet Le & Yang (2015), Flower102 Gogul & Kumar (2017), and one NLP dataset: AGNews. These represent varying complexity levels and inter-class similarity. For statistical heterogeneity evaluation, we use the Dirichlet distribution with parameter $\alpha$ to partition data among clients.

**Model Architectures.** To simulate architectural heterogeneity, we utilize eight diverse model architectures ranging from CNNs to ResNets and MobileNets for CV tasks (Appendix E, Table 7), and five Transformer architectures for NLP tasks (Appendix E, Table 8). Models are assigned cyclically from this pool. For homogeneous baselines (HmFL), all clients use Model_1.

**Baselines.** We compare FedFuse against two groups of baselines: *HmFL Baselines (adapted for personalization):* FedAvg McMahan et al. (2017), Per-FedAvg Fallah et al. (2020), FedProx Yuan & Li (2022), FedPer Arivazhagan et al. (2019). These are evaluated in the homogeneous setting to assess personalization capability without architectural heterogeneity. *HeteroFL Baselines:* FedKD Jeong et al. (2018), FedProto Tan et al. (2022), FedMRL Yi et al. (2024a), FedTGP Zhang et al. (2024). These methods are designed to handle architectural and statistical heterogeneity.

**Implementation Details.** All experiments are implemented using PyTorch 2.1.0 and conducted on NVIDIA 4090D GPUs. Key hyperparameters (learning rates, epochs, batch sizes, MoE configuration, etc.) are detailed in Appendix E, Subsection E.2. Unless otherwise specified, results are averaged over 3 runs with different random seeds. The reported accuracy is the average test accuracy across all participating clients on their respective local test sets after the final communication round.

### 4.1 PERFORMANCE EVALUATION

#### 4.1.1 PERSONALIZED ACCURACY COMPARISON

We first compare the final personalized test accuracy of FedFuse against baselines under both HmFL and HeteroFL settings across varying client numbers ($K = 10, 50, 100, 500$). The results are summarized in Table 1 (for $K = 10$) and Tables 9, 10 in Appendix E (for $K = 50, 100, 500$).

FedFuse consistently achieves higher average test accuracy compared to all baseline methods across the three datasets and client scales in the HeteroFL setting. For instance, in the 10-client HeteroFL scenario (Table 1), FedFuse surpasses the best performing baseline by substantial margins: ↑11.1% on CIFAR-100, ↑46.7% on Tiny-Imagenet, and ↑26.7% on Flower102. Similar significant gains are observed for $K = 50$ and $K = 100$. Moreover, FedFuse's advantage grows with data complexity. For example, on CIFAR-100, FedFuse outperforms the second-best method by about 10%; on Tiny-Imagenet, the improvement ranges from 20% to 40%; and on Flower102, it ranges from 10% to 40%. This stems from FedFuse's use of the Mixture-of-Experts (MoE) architecture to decouple knowledge rather than simply aggregating it. By doing so, the model can fully absorb knowledge from different clients, giving FedFuse a significant edge in complex tasks. We also evaluate the performance on NLP datasets, where FedFuse achieves a significant improvement of 23.7%. Detailed experimental results can be found in the Appendix (Table 11).

Table 1: Comparison on three datasets with 10 clients. The best result is **bold**, the second is underlined.

| Acc(%) | Algorithm | CIFAR-100 | Tiny-Imagenet | Flower102 |
|---|---|---|---|---|
| | FedAvg McMahan et al. (2017) | 31.76 | 15.82 | 20.18 |
| | Per-FedAvg Fallah et al. (2020) | 31.57 | 26.30 | 22.33 |
| **HmFL** | FedProx Yuan & Li (2022) | 31.78 | 16.70 | 19.06 |
| | FedPer Arivazhagan et al. (2019) | 39.94 | 31.12 | 44.31 |
| | **FedFuse** | **43.71** (↑9.4%) | **33.64** (↑8.1%) | **49.16** (↑10.9%) |
| | FedKD Jeong et al. (2018) | 35.90 | 21.95 | 39.24 |
| | FedProto Tan et al. (2022) | 33.20 | 16.93 | 27.05 |
| **HeteroFL** | FedMRL Yi et al. (2024a) | 37.98 | 22.10 | 37.68 |
| | FedTGP Zhang et al. (2024) | 32.71 | 20.34 | 41.39 |
| | **FedFuse** | **42.22** (↑11.1%) | **32.43** (↑46.7%) | **52.46** (↑26.7%) |

Table 2: Evaluation under non-IID Data (Dirichlet $\alpha$). The best result is **bold**, the second is underlined.

| Acc(%) | Algorithm | Cifar-100 | | | Flower102 | | |
|---|---|---|---|---|---|---|---|
| | | $\alpha$=0.05 | $\alpha$=0.1 | $\alpha$=0.5 | $\alpha$=0.05 | $\alpha$=0.1 | $\alpha$=0.5 |
| | FedKD Jeong et al. (2018) | 26.36 | 16.75 | 10.88 | 21.74 | 18.42 | 9.5 |
| **HeteroFL** | FedProto Tan et al. (2022) | 19.85 | 18.75 | 11.18 | 23.81 | 19.90 | 12.53 |
| **(100 clients)** | FedMRL Yi et al. (2024a) | 36.28 | 23.97 | 14.01 | 20.95 | 26.87 | 13.07 |
| | FedTGP Zhang et al. (2024) | 26.53 | 16.70 | 11.11 | 23.68 | 21.78 | 13.85 |
| | **Ours** | **45.22** | **31.90** | **21.03** | **43.85** | **30.56** | **19.06** |
| | FedKD Jeong et al. (2018) | 35.86 | 30.37 | 10.24 | 17.82 | 26.71 | 12.93 |
| **HeteroFL** | FedProto Tan et al. (2022) | 33.62 | 25.61 | 13.39 | 24.65 | 12.74 | 14.30 |
| **(50 clients)** | FedMRL Yi et al. (2024a) | 40.09 | 30.91 | 12.76 | 20.44 | 26.32 | 12.90 |
| | FedTGP Zhang et al. (2024) | 31.14 | 25.93 | 13.64 | 12.59 | 27.87 | 14.33 |
| | **Ours** | **44.32** | **36.53** | **21.52** | **53.98** | **39.48** | **23.67** |
| | FedKD Jeong et al. (2018) | 41.38 | 35.90 | 23.35 | 40.73 | 39.24 | 22.91 |
| **HeteroFL** | FedProto Tan et al. (2022) | 33.09 | 33.20 | 17.12 | 28.39 | 27.05 | 11.65 |
| **(10 clients)** | FedMRL Yi et al. (2024a) | 45.30 | 37.98 | 24.06 | 39.26 | 37.68 | 22.23 |
| | FedTGP Zhang et al. (2024) | 40.88 | 32.71 | 20.28 | 47.36 | 41.39 | 26.08 |
| | **Ours** | **47.70** | **42.22** | **27.70** | **61.27** | **52.46** | **36.07** |

### 4.1.2 CONVERGENCE SPEED

Figure 5 presents the training curves (average test accuracy vs. communication rounds) for the HeteroFL setting. FedFuse generally demonstrates faster convergence compared to baselines, exhibiting a rapid accuracy increase in the early training stages, followed by steady growth. It consistently outperforms other methods throughout the training process, with its advantage often becoming more pronounced in the later stages. For instance, observing the curves in Figure 5, on the Tiny-Imagenet dataset with 50 clients (Figure 5e), FedFuse's accuracy is significantly higher than others by the middle of training and stabilizes later. Similarly, on the Flower102 dataset with 100 clients (Figure 5i), it maintains the highest accuracy throughout, with its lead over other methods widening over time. Overall, FedFuse exhibits excellent training speed and convergence properties in the HeteroFL setting. The corresponding curves for the HmFL setting, showing similar trends of fast convergence for FedFuse, are provided in Figure 6 (Appendix E).

### 4.1.3 ROBUSTNESS TO DATA HETEROGENEITY

We evaluate the impact of statistical heterogeneity using the Dirichlet distribution Dir($\alpha$) with $\alpha \in \{0.5, 0.1, 0.05\}$. Table 2 shows the performance under these conditions for $K = 10, 50, 100$. FedFuse consistently achieves the highest accuracy across all settings, demonstrating superior robustness. For instance, with 100 clients, FedFuse achieves the highest accuracy of 45.22% on the CIFAR-100 dataset [for $\alpha = 0.05$] and 43.85% on the Flower102 dataset [for $\alpha = 0.05$], both markedly outperforming other algorithms. This suggests that FedFuse possesses enhanced robustness and superior performance when dealing with non-IID data. Its relative advantage often widens under higher heterogeneity (e.g., $\alpha = 0.05$), supporting the hypothesis that adaptive MoE aggregation is particularly beneficial when client data differs significantly.

### 4.2 ABLATION STUDIES

To dissect the contribution of the key components of FedFuse, we perform ablation studies on CIFAR-100 with K=50, $\alpha = 0.1$. We compare the full FedFuse framework against several variants:

**w/o MoE:** Replaces MoE aggregation with simple averaging of client logits, but keeps the $\mathcal{L}_p$ update.

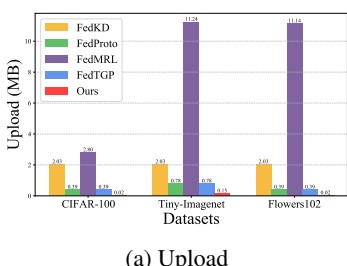

(a) Upload

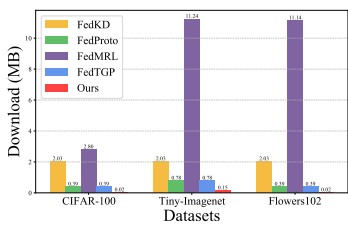

(b) Download

Figure 2: Communication Overhead with 10-Client HeteroFL. Illustrates model-size independence.

**w/o $\mathcal{L}_p$:** Uses MoE aggregation, but removes the final personalization update step (Equation 16).

**FedGen** Zhu et al. (2021b): A baseline employs a global generator to aggregate logits and cross-entropy loss for client updates (no MoE, no specific $\mathcal{L}_p$).

The results in Table 3 demonstrate the importance of both components. Removing either MoE or the $\mathcal{L}_p$ update results in a noticeable performance drop of 0.56% and 7.2% percentage points, respectively, compared to the full FedFuse. Both components are necessary to achieve the best performance, outperforming the simpler LogitsAvg-KD (FedGen) baseline significantly. This validates the design choices of using adaptive MoE for aggregation and the specific reverse KL loss for personalization.

Table 3: Ablation study on CIFAR100, 50 clients.

| Method Variant $\alpha = 0.1$. | Accuracy (%) |
|---|---|
| FedFuse (Full) | **36.53** |
| w/o MoE Aggregation | 35.97 |
| w/o Personalized Update $\mathcal{L}_p$ | 29.33 |
| FedGen Zhu et al. (2021b) | 14.92 |
| FedKD Jeong et al. (2018) | 30.37 |

Table 4: Computation study on CIFAR100

| Method | Avg. Client Time (s) | Server Time (s) |
|---|---|---|
| FedFuse | 1.98 | 1.76 |
| FedKD | 5.15 | 11.01 |
| FedProto | 1.18 | 40.33 |
| FedTGP | 1.56 | 42.87 |
| FedMRL | 1.16 | 8.56 |

Table 5: Time Complexity Analysis

| Algorithm | FedKD | FedProto | FedTGP | FedMRL | Ours |
|---|---|---|---|---|---|
| **Complexity** | $O(N \times C)$ | $O(N \times C)$ | $O(N \times C^2)$ | $O(N \times C)$ | $O(N \times C)$ |

## 4.3 RESOURCE OVERHEAD ANALYSIS

We analyze the resource usage of FedFuse compared to baselines. Table 5 summarizes the communication time complexities, with detailed computational complexity analysis provided in Appendix C.1.

**Communication Overhead.** We measure the total data transferred per round (client uploads + server downloads) in Megabytes (MB). Since FedFuse only transmits logits, its communication cost is independent of client model sizes. Figure 2 illustrates the upload cost specifically for K=10. Compared to other knowledge-fusion methods, FedFuse's communication is determined by $|\mathcal{D}^b| \times C$. We find that FedFuse requires lower communication volume than FedMRL and FedTGP.

**Computation Overhead.** We measure the average wall-clock time per round with 50 clients and $\alpha = 0.1$. Table 4 reports the average time from the client and the server side. Client time heavily depends on $E_c$ and local model complexity $L_i$. Server time depends on $E_s$ and MoE complexity. We observe that FedFuse's server time is 1.76s, lower than the baselines.

## 5 CONCLUSION

In this work, we introduce FedFuse, a novel framework designed to tackle the critical issues of personalized knowledge loss in HeteroFL. By uniquely combining an Expert-guided Fusion mechanism for adaptive knowledge aggregation with a selective knowledge distillation strategy for preserving personalization, FedFuse effectively addresses the limitations of prior methods. Extensive experiments confirmed FedFuse's significant accuracy improvements over state-of-the-art HeteroFL baselines, particularly under high heterogeneity, while demonstrating favorable resource trade-offs. While FedFuse's effectiveness has been demonstrated in near-real-world settings, practical deployment on physical devices and in environments with extremely large models remains untested due to resource constraints. We will conduct practical deployment in the future work.

## ETHICS STATEMENT

This work does not involve human subjects, sensitive data, or clear immediate paths to deployment in high-risk applications. Therefore, we do not foresee significant ethical issues arising directly from this work.

## REPRODUCIBILITY STATEMENT

To facilitate the reproducibility of our work, we have made the following efforts: 1) Detailed hyperparameter settings and experimental configurations for each figure and table are provided in Appendix E.2. 2) We use publicly available datasets (e.g., CIFAR-100, Tiny-ImageNet, Flower102 and AGNews).

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

# Appendix

## USE OF LLM

We used a Large Language Model (LLM) for grammar proofreading and language polishing in this paper.

## A  ALGORITHM

---

**Algorithm 2** FedFuse: Selective Knowledge Distillation with Expert-Guided Fusion

---

**Require:** Communication rounds $T$, client participation fraction $p$, private datasets $\{\mathcal{D}_i^r\}_{i=1}^K$, public dataset $\mathcal{D}^b$, initial client models $\{\theta_i^0\}_{i=1}^K$, initial global model $\theta_g^0$, local epochs $E_c$, server epochs $E_s$, learning rates $\eta, \eta_g, \eta_p$, temperature $\tau$.

1: **for** each round $t = 1, 2, \ldots, T$ **do**
2:    Select a subset of clients $N_t \subseteq \{1, \ldots, K\}, |N_t| = \max(1, \lfloor p \cdot K \rfloor)$
3:    **Client Local Training and Knowledge Representation (in parallel for $i \in N_t$):**
4:    Let $\theta_i' \leftarrow \theta_i^{t-1}$
5:    **for** local epoch $e = 1, \ldots, E_c$ **do**
6:       Update $\theta_i'$ using Equation 3 on $\mathcal{D}_i^r$
7:    **end for**
8:    Compute local logits $c_i^t = \{c_i^t(x) | x \in \mathcal{D}^b\}$ via Equation 4 and Equation 5 using $\theta_i'$
9:    Send $c_i^t$ to server
10:   Let $\theta_i^t \leftarrow \theta_i'$
11:   **Call Algorithm 1** with $\theta_g^{t-1}, \{c_i^t\}_{i \in N_t}, \mathcal{D}^b, E_s, \eta_g$ to get $\tilde{c}_g^t$ and $\theta_g^t$
12:   Distribute $\tilde{c}_g^t$ to clients in $N_t$
13:   **Selective Knowledge Distillation (in parallel for $i \in N_t$):**
14:   Receive $\tilde{c}_g^t$
15:   Compute personalized loss $\mathcal{L}_p$ via Equation 15 using $\tilde{c}_g^t$ and logits from current $\theta_i^t$
16:   Update local model via Equation 16: $\theta_i^t \leftarrow \theta_i^t - \eta_p \nabla_{\theta_i^t} \mathcal{L}_p$
17: **end for**
18: **return** Final personalized client models $\{\theta_i^T\}_{i=1}^K$ (or $\{\theta_i^T\}_{i \in N_T}$ if only a subset has the final update)

---

## B  NOTATION

Table 6 summarizes the main notations used throughout the paper.

## C  THEORETICAL ANALYSIS

This section provides supplementary details for the theoretical analysis discussed in Section 3.6.

### C.1  COMPLEXITY ANALYSIS DETAILS

As mentioned in the main text, the time complexity of FedFuse in each communication round $t$ involves several components:

1. **Client-Side Computation:**

- **Local Training:** Each of the $|N_t|$ participating clients performs $E_c$ local epochs. Within each epoch, it processes $|\mathcal{D}_i^r|$ samples. Let $L_i$ be the average complexity (forward + backward pass) for one sample on client $i$'s model $M_i$. The total complexity for local training across selected clients is approximately $O(|N_t| \cdot E_c \cdot \max_i(|\mathcal{D}_i^r| \cdot L_i))$.

Table 6: Notations

| Symbol | Description |
|---|---|
| $K$ | Total number of clients. |
| $i$ | Index for clients. |
| $N_t$ | Set of clients participating in round $t$. |
| $T$ | Total number of communication rounds. |
| $\mathcal{D}_i^r$ | The client $i$'s private dataset. |
| $\mathcal{D}^b$ | The public dataset shared across clients and server. |
| $M_i, \theta_i$ | Local model and its parameters for client $i$. |
| $M_g, \theta_g$ | Global MoE model and its parameters on the server. |
| $f_{\theta_i}(\cdot)$ | Forward pass function for client model $i$. |
| $\psi_{\theta_g}(\cdot)$ | Feature extractor part of the global model $M_g$. |
| $C$ | Number of classes in the classification task (dimension of logits). |
| $c_{i,l}^t(x)$ | Raw output logits from client $i$ for input $x$ at round $t$. |
| $c_i^t$ | Temperature-scaled logits (probability distribution) from client $i$ for input $x$ at round $t$. |
| $c_g^t$ | Aggregated global logits from server MoE model for input $x$ at round $t$. |
| $\tilde{c}_g^t$ | Updated global logits distributed from server to clients at round $t$. |
| $\tau$ | Temperature parameter for scaling logits. |
| $\eta, \eta_g, \eta_p$ | Learning rates for local training, server aggregation, and local personalization update. |
| $\lambda$ | L2 regularization coefficient for local training. |
| $E_c$ | Number of local training epochs per round. |
| $E_s$ | Number of server training epochs per round. |
| $E$ | The total number of experts in the MoE layer. |
| $k$ | The number of active experts selected by the Top-k gating mechanism. |
| $\mathcal{E}(x)$ | Set of indices for the top-k active experts for input $x$. |
| $\pi_j(\cdot)$ | Gating weight for expert $j$. |
| $e_j(\cdot)$ | Output function for expert $j$. |
| $d_{in}$ | Input dimension for the MoE layer (output dimension of $\psi$). |
| $L_i$ | Computational complexity of one forward/backward pass for client $i$. |
| $L_g^e$ | Computational complexity of the global model's feature extractor $\psi$. |
| $\mathcal{L}_c$ | Cross-Entropy loss function. |
| $\mathcal{L}_a$ | Server-side aggregation loss (based on KL divergence). |
| $\mathcal{L}_p$ | Client-side personalization loss (based on reverse KL divergence). |
| $D_K(P \parallel Q)$ | Kullback-Leibler divergence from distribution P to Q. |

- **Logits Generation:** Each client computes logits on the public dataset $\mathcal{D}^b$. This involves one forward pass per sample. Complexity is $O(|N_t| \cdot |\mathcal{D}^b| \cdot L_i^{fwd})$, where $L_i^{fwd}$ is the forward pass complexity.

- **Personalization Update:** Each client computes the loss $\mathcal{L}_p$ and performs one gradient update. This involves one forward pass on $\mathcal{D}^b$ and one backward pass. Complexity is $O(|N_t| \cdot |\mathcal{D}^b| \cdot L_i)$.

2. **Server-Side Computation:**

- **Aggregation Training:** The server performs $E_s$ epochs to update the global MoE model. In each epoch, it processes $|\mathcal{D}^b|$ samples. Let $L_g$ be the complexity of the global model ($M_g$) pass. $L_g$ includes the feature extractor ($\psi$, complexity $L_g^e$) and the MoE layer. The MoE layer involves computing gating weights ($O(d_{in}E)$), selecting top-k experts, and computing weighted expert outputs ($O(kCd_{in})$ for linear experts). The backward pass has similar complexity. Total server training complexity is $O(E_s \cdot |\mathcal{D}^b| \cdot L_g)$.

- **Global Logits Generation:** Computing $\tilde{c}_g^t$ involves one forward pass over $\mathcal{D}^b$, complexity $O(|\mathcal{D}^b| \cdot L_g^{fwd})$.

3. **Communication:**

- **Upload:** $|N_t|$ clients upload logits for $|\mathcal{D}^b|$ samples, dimension $C$. Total size $O(|N_t| \cdot |\mathcal{D}^b| \cdot C)$.
- **Download:** Server broadcasts global logits $\tilde{c}_g^t$ to $|N_t|$ clients. Total size $O(|N_t| \cdot |\mathcal{D}^b| \cdot C)$.

Assuming $L_i \approx L_i^{fwd} + L_i^{bwd}$ and $L_g \approx L_g^{fwd} + L_g^{bwd}$, and often $E_c \gg 1, E_s \geq 1$. The overall per-round complexity depends heavily on the relative sizes of datasets, local vs server epochs, and model complexities. The simplified complexity $O(N \times C)$ mentioned in the original draft likely refers only to the communication cost under specific assumptions and neglects computational costs, which can be substantial, especially local training.

## C.2 CONVERGENCE ANALYSIS OF FEDFUSE

In this subsection, we provide a formal convergence guarantee for the proposed FedFuse framework under standard assumptions commonly used in federated optimization.

**Assumption 1** (Smoothness). *$F$ is $L$-smooth: for all vector $\theta_1$ and $\theta_2$,*

$$\|\nabla F(\theta_1) - \nabla F(\theta_2)\| \leq L\|\theta_1 - \theta_2\|.$$

*Another form:*

$$F(\theta_1) \leq F(\theta_2) + <\theta_1 - \theta_2, \nabla F(\theta_2)> + \frac{L}{2}\|\theta_1 - \theta_2\|^2.$$

**Assumption 2** (Convexity). *Assume $F$ is strongly convex with parameter $\mu$. For any vector $\theta_1$, $\theta_2$, we have:*

$$F(\theta_1) \geq F(\theta_2) + <\theta_1 - \theta_2, \nabla F(\theta_2)> + \frac{\mu}{2}\|\theta_1 - \theta_2\|^2.$$

**Assumption 3** (Stochastic Gradient Variance Bounded).

$$E[\|\nabla F(\theta; x, y) - \nabla F(\theta)\|^2] \leq \sigma^2.$$

**Assumption 4** (MoE Boundedness). *The gating weights $\{\pi_j\}$ sum to 1 (over the selected $k$ experts), and each expert's outputs and gradients are uniformly bounded.*

## C.3 KEY LEMMAS

We establish several lemmas to facilitate the convergence proofs.

**Lemma 1.** *Assume Assumption 1, 2, 3 hold, if learning rate $\eta \leq \frac{2\mu^2}{L^3}$. Then the local update on the client side satisfies:*

$$E[L_c(\theta_i^{t+1})] \leq E[L_c(\theta_i^t)] - \mu^2 \eta E[\|\theta_i^t - \theta_i^*\|^2] + \frac{\eta^2 L \sigma^2}{2}. \tag{19}$$

**Lemma 2.** *Assume Assumption 1, 2, 3, 4 hold, if learning rate $\eta_g \leq \frac{2\mu^2}{\beta^3}$. Then:*

$$E[L_a(\theta_g^{t+1})] \leq E[L_a(\theta_g^t)] - \mu^2 \eta_g E[\|\theta_g^t - \theta_g^*\|^2] + \frac{\eta_g^2 \beta \sigma^2}{2}. \tag{20}$$

**Lemma 3.** *Assume Assumption 1, 2, 3, when $\eta_p \leq \frac{2}{\gamma}$,*

$$E[\|\tilde{\theta}_i^{t+1}\|] \leq E[L_p(\tilde{\theta}_i^t)] - \mu^2 \eta_p E[\|\tilde{\theta}_i^t - \tilde{\theta}_i^*\|^2] + \frac{\gamma \eta_p^2 \sigma^2}{2}. \tag{21}$$

**Lemma 4.** *Assume Assumption 1 and 3, it follows:*

$$E[\nabla L_c(\theta_i^{t+1})] \leq E[\nabla L_c(\theta_i^t)] - \eta(1 - \frac{\eta L}{2})E[\|\nabla L_c(\theta_i^t)\|^2] + \frac{\eta^2 L \sigma^2}{2}. \tag{22}$$

**Lemma 5.** *Assume Assumption 1, 3, 4, it follows:*

$$E[\nabla L_p(\tilde{\theta}_i^{t+1})] \leq E[\nabla L_p(\tilde{\theta}_i^t)] - \eta_p(1 - \frac{\eta_p L}{2})E[\|\nabla L_p(\tilde{\theta}_i^t)\|^2] + \frac{\eta_p^2 L \sigma^2}{2}. \tag{23}$$

**Lemma 6.** *Assume Assumption 1, 3, it follows:*

$$E[\nabla L_a(\theta_g^{t+1})] \leq E[\nabla L_a(\theta_g^t)] - \eta_g(1 - \frac{\eta_g L}{2})E[\|\nabla L_a(\theta_g^t)\|^2] + \frac{\eta_g^2 L \sigma^2}{2}. \tag{24}$$

## C.4 CONVERGENCE ANALYSIS FOR STRONGLY CONVEX CASE

*Proof.* We start by defining the combined loss function as:

$$F(\theta) = L_c(\theta) + L_a(\theta) + L_p(\theta). \tag{25}$$

Based on Lemma 1, Lemma 2, and Lemma 3, we can infer that the learning rates are equal, i.e., $\eta = \eta_p = \eta_g$. This allows us to combine the results from the individual lemmas.

First, we consider the sum of the expected differences in the loss function over all iterations:

$$\sum_{t=1}^{T} E[F(\theta^t) - F(\theta^{t+1})] \geq \mu^2 \eta \sum_{t=1}^{T} E[\|\theta_i^t - \theta_i^*\|^2 + \|\theta_g^t - \theta_g^*\|^2 + \|\tilde{\theta}_i^t - \tilde{\theta}_i^*\|^2] - \frac{\eta^2 \sigma^2 T}{2}(L + \beta + \gamma). \tag{26}$$

This inequality provides a lower bound on the sum of the expected differences in the loss function.

Next, we observe that the sum of the expected differences can also be written as:

$$\sum_{t=1}^{T} E[F(\theta^t) - F(\theta^{t+1})] = E[F(\theta^1) - F(\theta^{T+1})] \leq F(\theta^1) - F(\theta^*). \tag{27}$$

This follows from the fact that the expected value of the loss function at the final iteration is less than or equal to the loss function at the initial iteration.

Combining the two inequalities, we obtain:

$$\mu^2 \eta \sum_{t=1}^{T} E[\|\theta_i^t - \theta_i^*\|^2 + \|\theta_g^t - \theta_g^*\|^2 + \|\tilde{\theta}_i^t - \tilde{\theta}_i^*\|^2] \leq F(\theta^1) - F(\theta^*) + \frac{\eta^2 \sigma^2 T}{2}(L + \beta + \gamma). \tag{28}$$

This inequality provides a bound on the sum of the squared differences between the current parameters and the optimal parameters.

Next, we use the strong convexity property of the individual loss functions to get:

$$\frac{\mu}{2}\|\theta_i^t - \theta_i^*\|^2 \leq L_c(\theta_i^t) - L_c(\theta_i^*). \tag{29}$$

$$\frac{\mu}{2}\|\tilde{\theta}_i^t - \tilde{\theta}_i^*\|^2 \leq L_p(\tilde{\theta}_i^t) - L_p(\tilde{\theta}_i^*). \tag{30}$$

$$\frac{\mu}{2}\|\theta_g^t - \theta_g^*\|^2 \leq L_a(\theta_g^t) - L_a(\theta_g^*). \tag{31}$$

Summing these inequalities, we get:

$$\frac{\mu}{2}[\|\theta_i^t - \theta_i^*\|^2 + \|\theta_g^t - \theta_g^*\|^2 + \|\tilde{\theta}_i^t - \tilde{\theta}_i^*\|^2] \leq F(\theta^t) - F(\theta^*). \tag{32}$$

This inequality provides a bound on the combined squared differences in terms of the combined loss function.

Using this result, we can bound the sum of the expected differences in the loss function as:

$$\sum_{t=1}^{T} E[F(\theta^t) - F(\theta^*)] \leq \frac{1}{2\mu\eta}[F(\theta^1) - F(\theta^*)] + \frac{\eta\sigma^2 T}{4\mu}(L + \beta + \gamma). \tag{33}$$

Finally, dividing both sides by $T$, we obtain the desired result:

$$E[F(\theta^t) - F(\theta^*)] \leq \frac{1}{2\mu\eta T}[F(\theta^1) - F(\theta^*)] + \frac{\eta\sigma^2}{4\mu}(L + \beta + \gamma). \tag{34}$$

This inequality provides a bound on the expected difference in the loss function at each iteration. $\square$

### C.4.1 PROOF OF LEMMA 1

*Proof.* Firstly, we leverage the strong convexity of $L_c$ with parameter $\mu$. This implies that the gradient norm has a lower bound as follows:

$$\|\nabla L_c(\theta_i^t)\| \geq \mu^2 \|\theta_i^t - \theta_i^*\|^2. \tag{35}$$

Next, by Assumption 3, we can get an upper bound for the expected squared gradient norm:

$$E[\|\nabla L(\theta; x, y)\|^2] \leq L^2 E[\|\theta - \theta^*\|] + \sigma^2. \tag{36}$$

Then, considering the update rule $\theta_i^{t+1} = \theta_i^t - \eta \nabla L_c(\theta_i^t)$ and the L-smoothness of $L_c$, we have the following inequality:

$$L_c(\theta_i^{t+1}) \leq L_c(\theta_i^t) + \langle \theta_i^{t+1} - \theta_i^t, \nabla L_c(\theta_i^t) \rangle + \frac{L}{2}\|\theta_i^{t+1} - \theta_i^t\|^2 \tag{37}$$

$$\leq L_c(\theta_i^t) - \eta\|\nabla L_c(\theta_i^t)\|^2 + \frac{\eta^2 L}{2}\|\nabla L_c(\theta_i^t)\|^2. \tag{38}$$

Taking the expectation on both sides, we obtain:

$$E[L_c(\theta_i^{t+1})] \leq E[L_c(\theta_i^t)] - \eta E[\|\nabla L_c(\theta_i^t)\|^2] + \frac{\eta^2 L}{2} E[\|\nabla L_c(\theta_i^t)\|^2] \tag{39}$$

$$\leq E[L_c(\theta_i^t)] - \mu^2 \eta E[\|\theta_i^t - \theta_i^*\|^2] + \frac{\eta^2 L \sigma^2}{2}. \tag{40}$$

To ensure the desired inequality holds, we need the coefficient of $E[\|\theta_i^t - \theta_i^*\|^2]$ to be non-negative, which leads to the condition $\mu^2 - \frac{L^3 \eta}{2} \geq 0$. Solving this inequality, we get $\eta \leq \frac{2\mu^2}{L^3}$.

$\square$

### C.4.2 PROOF OF LEMMA 2

*Proof.* We start by considering the gradient of the aggregated loss $L_a$ for the server parameters $\theta_g$. It is known that:

$$E[\|\nabla L_a\|^2] \leq \beta^2 E[\|\theta_g^t - \theta_g^*\|^2] + \sigma^2. \tag{41}$$

This inequality provides an upper bound for the expected squared gradient norm of the aggregated loss.

Next, we analyze the update rule for $\theta_g$. Given the update rule $\theta_g^{t+1} = \theta_g^t - \eta_g \nabla L_a(\theta_g^t)$ and the $\beta$-smoothness of $L_a$, we have:

$$L_a(\theta_g^{t+1}) \leq L_a(\theta_g^t) + \langle \theta_g^{t+1} - \theta_g^t, \nabla L_a(\theta_g^t) \rangle + \frac{\beta}{2}\|\theta_g^{t+1} - \theta_g^t\|^2 \tag{42}$$

$$\leq L_a(\theta_g^t) - \eta_g\|\nabla L_a(\theta_g^t)\|^2 + \frac{\eta_g^2 \beta}{2}\|\nabla L_a(\theta_g^t)\|^2. \tag{43}$$

Taking the expectation on both sides, we obtain:

$$E[L_a(\theta_g^{t+1})] \leq E[L_a(\theta_g^t)] - \eta_g E[\|\nabla L_a(\theta_g^t)\|^2] + \frac{\eta_g^2 \beta}{2} E[\|\nabla L_a(\theta_g^t)\|^2] \tag{44}$$

$$\leq E[L_a(\theta_g^t)] - \mu^2 \eta_g E[\|\theta_g^t - \theta_g^*\|^2] + \frac{\eta_g^2 \beta \sigma^2}{2}. \tag{45}$$

To ensure the desired inequality holds, we need the coefficient of $E[\|\theta_g^t - \theta_g^*\|^2]$ to be non-negative. This leads to the condition $\mu^2 - \frac{\beta^3 \eta_g}{2} \geq 0$. Solving this inequality, we get $\eta_g \leq \frac{2\mu^2}{\beta^3}$. $\square$

### C.4.3 PROOF OF LEMMA 3

*Proof.* We begin by applying the update rule and the smoothness property of the loss function. Specifically, for the local update on the client side, we have:

$$L_p(\tilde{\theta}_i^{t+1}) \leq L_p(\tilde{\theta}_i^t) + \langle \tilde{\theta}_i^{t+1} - \tilde{\theta}_i^t, \nabla L_p(\tilde{\theta}_i^t) \rangle + \frac{\gamma}{2}\|\tilde{\theta}_i^{t+1} - \tilde{\theta}_i^t\|^2 \tag{46}$$

$$\leq L_p(\tilde{\theta}_i^t) - \eta_p\|\nabla L_p(\tilde{\theta}_i^t)\|^2 + \frac{\eta_p^2 \gamma}{2}\|\nabla L_p(\tilde{\theta}_i^t)\|^2. \tag{47}$$

Here, the first inequality follows from the $\gamma$-smoothness of $L_p$, and the second inequality follows from the update rule $\tilde{\theta}_i^{t+1} = \tilde{\theta}_i^t - \eta_p \nabla L_p(\tilde{\theta}_i^t)$.

Next, we use the strong convexity property of the loss function, which implies that:

$$\|\nabla L_p(\tilde{\theta}_i^t)\|^2 \geq \mu^2 \|\tilde{\theta}_i^t - \tilde{\theta}_i^*\|^2. \tag{48}$$

Taking the expectation on both sides of the inequalities in Equation(47) and using the result from Equation(48), we get:

$$E[L_p(\tilde{\theta}_i^{t+1})] \leq E[L_p(\tilde{\theta}_i^t)] - \eta_p E[\|\nabla L_p(\tilde{\theta}_i^t)\|^2] + \frac{\eta_p^2 \gamma}{2} E[\|\nabla L_p(\tilde{\theta}_i^t)\|^2] \tag{49}$$

$$\leq E[L_p(\tilde{\theta}_i^t)] - \mu^2 \eta_p E[\|\tilde{\theta}_i^t - \tilde{\theta}_i^*\|^2] + \frac{\eta_p^2 \gamma \sigma^2}{2}. \tag{50}$$

To ensure the desired inequality holds, we need the coefficient of $E[\|\tilde{\theta}_i^t - \tilde{\theta}_i^*\|^2]$ to be non-negative. This leads to the condition $\mu^2 - \frac{\mu^2 \gamma \eta_p}{2} \geq 0$. Solving this inequality, we obtain $\eta_p \leq \frac{2}{\gamma}$.

$\square$

### C.5   CONVERGENCE ANALYSIS FOR NON-CONVEX CASE

*Proof.* We start by defining the total loss function as the sum of the client, global, and personal losses:

$$F(\theta) = L_c(\theta) + L_a(\theta) + L_p(\theta). \tag{51}$$

Based on Lemma 4, Lemma 5, and Lemma 6, we can infer that the learning rates are equal, i.e., $\eta = \eta_p = \eta_g$.

Next, we combine the results from the individual lemmas to derive an inequality for the total loss function:

$$E[F(\theta^{t+1})] \leq E[F(\theta^t)] - \eta(1 - \frac{\eta L}{2}) E[\|\nabla L_c(\theta_i^t)\|^2 + \|\nabla L_a(\theta_g^t)\|^2 + \|\nabla L_p(\tilde{\theta}_i^t)\|^2] + \frac{3\eta^2 L \sigma^2}{2}. \tag{52}$$

This inequality provides a bound on the expected decrease in the total loss function at each iteration.

Summing this inequality over all iterations, we get:

$$\sum_{t=1}^{T} E[F(\theta^t) - F(\theta^{t+1})] \geq \eta(1 - \frac{\eta L}{2}) \sum_{t=1}^{T} E[\|\nabla F(\theta^t)\|^2] - \frac{3T\eta^2 L \sigma^2}{2}. \tag{53}$$

This inequality provides a lower bound on the sum of the expected differences in the total loss function over all iterations.

Rearranging the terms, we obtain:

$$\sum_{t=1}^{T} E[\|F(\theta^t)\|^2] \leq \frac{F(\theta^1) - F(\theta^*)}{\eta(1 - \frac{\eta L}{2})} + \frac{3T\eta L \sigma^2}{2(1 - \frac{\eta L}{2})} \tag{54}$$

$$\leq \frac{2}{\eta(1 - \eta L)}(F(\theta^1) - F(\theta^*)) + \frac{3T\eta L \sigma^2}{2 - \eta L}. \tag{55}$$

Finally, dividing both sides by $T$, we get the desired result:

$$E[\|F(\theta^t)\|^2] \leq \frac{2}{\eta T(1 - \eta L)}(F(\theta^1) - F(\theta^*)) + \frac{3\eta L \sigma^2}{2 - \eta L}. \tag{56}$$

This inequality provides a bound on the expected squared norm of the total loss function at each iteration. $\square$

### C.5.1   PROOF OF LEMMA 4

*Proof.* We start by applying the update rule and the smoothness property of the loss function. Specifically, for the local update on the client side, we have:

$$L_c(\theta_i^{t+1}) \leq L_c(\theta_i^t) + \langle \nabla L_c(\theta_i^t), \theta_i^{t+1} - \theta_i^t \rangle + \frac{L}{2} \|\theta_i^{t+1} - \theta_i^t\|^2. \tag{57}$$

Given the update rule $\theta_i^{t+1} = \theta_i^t - \eta\nabla L_c(\theta_i^t; x, y)$, we substitute this into the above inequality:

$$L_c(\theta_i^{t+1}) \leq L_c(\theta_i^t) + \eta\langle\nabla L_c(\theta_i^t), \nabla L_c(\theta_i^t; x, y)\rangle + \frac{\eta^2 L}{2}\|\nabla L_c(\theta_i^t; x, y)\|^2. \tag{58}$$

Next, we analyze the expected squared gradient norm. By the definition of the gradient and the noise term, we have:

$$E[\|\nabla L_c(\theta_i^t; x, y)\|^2] = E[\|\nabla L_c(\theta_i^t)\|^2] + E[\|\nabla L_c(\theta_i^t; x, y)\| - \|\nabla L_c(\theta_i^t)\|^2] \tag{59}$$

$$\leq E[\|\nabla L_c(\theta_i^t)\|^2] + \sigma^2. \tag{60}$$

Taking the expectation on both sides of the inequality in Equation(58) and using the result from Equation(60), we get:

$$E[\nabla L_c(\theta_i^{t+1})] \leq E[L_c(\theta_i^t)] - \eta E[\|\nabla L_c(\theta_i^t)\|^2] + \frac{\eta^2 L}{2}(E[\|\nabla L_c(\theta_i^t)\|^2] + \sigma^2) \tag{61}$$

$$\leq E[\nabla L_c(\theta_i^t)] - \eta(1 - \frac{\eta L}{2})E[\|\nabla L_c(\theta_i^t)\|^2] + \frac{\eta^2 L\sigma^2}{2}. \tag{62}$$

$\square$

### C.5.2 PROOF OF LEMMA 5

*Proof.* The proof for Lemma 5 follows a similar structure to Lemma 4, focusing on the personal loss function $L_p$. We start by applying the update rule and the smoothness property of the loss function. Specifically, for the personal update on the client side, we have:

$$L_p(\tilde{\theta}_i^{t+1}) \leq L_p(\tilde{\theta}_i^t) + \langle\nabla L_p(\tilde{\theta}_i^t), \tilde{\theta}_i^{t+1} - \tilde{\theta}_i^t\rangle + \frac{L}{2}\|\tilde{\theta}_i^{t+1} - \tilde{\theta}_i^t\|^2. \tag{63}$$

Given the update rule $\tilde{\theta}_i^{t+1} = \tilde{\theta}_i^t - \eta_p\nabla L_p(\tilde{\theta}_i^t; x, y)$, we substitute this into the above inequality:

$$L_p(\tilde{\theta}_i^{t+1}) \leq L_p(\tilde{\theta}_i^t) + \eta_p\langle\nabla L_p(\tilde{\theta}_i^t), \nabla L_p(\tilde{\theta}_i^t; x, y)\rangle + \frac{\eta_p^2 L}{2}\|\nabla L_p(\tilde{\theta}_i^t; x, y)\|^2. \tag{64}$$

Next, we analyze the expected squared gradient norm. By the definition of the gradient and the noise term, we have:

$$E[\|\nabla L_p(\tilde{\theta}_i^t; x, y)\|^2] = E[\|\nabla L_p(\tilde{\theta}_i^t)\|^2] + E[\|\nabla L_p(\tilde{\theta}_i^t; x, y)\| - \|\nabla L_p(\tilde{\theta}_i^t)\|^2] \tag{65}$$

$$\leq E[\|\nabla L_p(\tilde{\theta}_i^t)\|^2] + \sigma^2. \tag{66}$$

Taking the expectation on both sides of the inequality in Equation(64) and using the result from Equation(66), we get:

$$E[\nabla L_p(\tilde{\theta}_i^{t+1})] \leq E[L_p(\tilde{\theta}_i^t)] - \eta_p E[\|\nabla L_p(\tilde{\theta}_i^t)\|^2] + \frac{\eta_p^2 L}{2}(E[\|\nabla L_p(\tilde{\theta}_i^t)\|^2] + \sigma^2) \tag{67}$$

$$\leq E[\nabla L_p(\tilde{\theta}_i^t)] - \eta_p(1 - \frac{\eta_p L}{2})E[\|\nabla L_p(\tilde{\theta}_i^t)\|^2] + \frac{\eta_p^2 L\sigma^2}{2}. \tag{68}$$

$\square$

### C.5.3 PROOF OF LEMMA 6

*Proof.* The proof for Lemma 6 focuses on the aggregated loss function $L_a$. We start by applying the update rule and the smoothness property of the loss function. Specifically, for the global update on the server side, we have:

$$L_a(\theta_g^{t+1}) \leq L_a(\theta_g^t) + \langle\nabla L_a(\theta_g^t), \theta_g^{t+1} - \theta_g^t\rangle + \frac{L}{2}\|\theta_g^{t+1} - \theta_g^t\|^2. \tag{69}$$

Given the update rule $\theta_g^{t+1} = \theta_g^t - \eta_g\nabla L_a(\theta_g^t; x, y)$, we substitute this into the above inequality:

$$L_a(\theta_g^{t+1}) \leq L_a(\theta_g^t) + \eta_g\langle\nabla L_a(\theta_g^t), \nabla L_a(\theta_g^t; x, y)\rangle + \frac{\eta_g^2 L}{2}\|\nabla L_a(\theta_g^t; x, y)\|^2. \tag{70}$$

Next, we analyze the expected squared gradient norm. By the definition of the gradient and the noise term, we have:

$$E[\|\nabla L_a(\theta_g^t; x, y)\|^2] = E[\|\nabla L_a(\theta_g^t)\|^2] + E[\|\nabla L_a(\theta_g^t; x, y)\| - \|\nabla L_a(\theta_g^t)\|^2] \quad (71)$$

$$\leq E[\|\nabla L_a(\theta_g^t)\|^2] + \sigma^2. \quad (72)$$

Taking the expectation on both sides of the inequality in Equation(70) and using the result from Equation(72), we get:

$$E[\nabla L_a(\theta_g^{t+1})] \leq E[L_a(\theta_g^t)] - \eta_g E[\|\nabla L_a(\theta_g^t)\|^2] + \frac{\eta_g^2 L}{2}(E[\|\nabla L_a(\theta_g^t)\|^2] + \sigma^2) \quad (73)$$

$$\leq E[\nabla L_a(\theta_g^t)] - \eta_g(1 - \frac{\eta_g L}{2})E[\|\nabla L_a(\theta_g^t)\|^2] + \frac{\eta_g^2 L \sigma^2}{2}. \quad (74)$$

$\square$

## D  DISCUSSION

In this work, we address the challenge of client heterogeneity in heterogeneous federated learning. To overcome these limitations, we propose FedFuse. FedFuse introduces a server-side Expert-guided Fusion mechanism that uniquely facilitates adaptive knowledge fusion by dynamically gating and weighting heterogeneous client knowledge contributions, moving beyond prior static schemes. Complementarily, an elaborately designed selective knowledge distillation strategy allows clients to assimilate global knowledge without blind imitation, thereby preserving crucial local model features and mitigating detrimental model divergence. The effectiveness of our approach is supported by theoretical analysis and extensive experiments conducted on various datasets and models for computer vision tasks.

The limitations still remain. While FedFuse's effectiveness has been demonstrated in near-real-world settings, practical deployment on physical devices and in environments with extremely large models remains untested due to resource constraints. Real-world implementation may uncover additional challenges or limitations, providing further insights into the system's scalability and efficiency.

## E  ADDITIONAL EXPERIMENTAL DETAILS

### E.1  MODEL SETTINGS

Under the model-heterogeneous (HeteroFL) setting, we employ eight distinct model architectures for collaborative training across clients, simulating diverse device capabilities. The specific architectures assigned to clients depend on the total number of clients $K$. For experiments with $K = 10, 50, 100$, models are assigned cyclically from the list below. In the model-homogeneous (HmFL) baseline experiments, all clients utilize the 'Model_1' architecture for fair comparison of personalization algorithms without architectural confounding. The detailed model configurations used in the HeteroFL setting are provided in Table 7.

Table 7: Model Configurations used in Heterogeneous Experiments for CV Tasks.

| Name | Base Architecture | Key Features | Layers | Params. |
|---|---|---|---|---|
| Model_1 | Simple CNN | 2 Conv layers, 3 FC layers | 5 | ≈3.2M |
| Model_2 | ResNet18 | BasicBlock, [2,2,2,2] layers | 18 | ≈11.2M |
| Model_3 | ResNet34 | BasicBlock, [3,4,6,3] layers | 34 | ≈21.3M |
| Model_4 | ResNet50 | Bottleneck, [3,4,6,3] layers | 50 | ≈23.5M |
| Model_5 | ResNet101 | Bottleneck, [3,4,23,3] layers | 101 | ≈42.5M |
| Model_6 | ResNet152 | Bottleneck, [3,8,36,3] layers | 152 | ≈58.2M |
| Model_7 | GoogleNet | Inception modules | 22 | ≈6.8M |
| Model_8 | MobileNetV2 | Linear Bottlenecks, Depthwise Separable Conv | 53 | ≈3.5M |
| Homo. | Simple CNN | 2 Conv layers, 3 FC layers | 5 | ≈3.2M |

Table 8: Model Configurations used in Heterogeneous Experiments for NLP Tasks.

| Name | Model Architecture | Key Features | Params. |
|---|---|---|---|
| Model_1 | Transformer | 1-head self-attention, 1 FFN | $\approx$4.12M |
| Model_2 | Transformer | 2-head self-attention, 2 FFN | $\approx$4.14M |
| Model_3 | Transformer | 4-head self-attention, 4 FFN | $\approx$4.18M |
| Model_4 | Transformer | 8-head self-attention, 8 FFN | $\approx$4.26M |
| Model_5 | Transformer | 16-head self-attention, 16 FFN | $\approx$4.42M |

Table 9: Comparison with the SOTA methods on three datasets in 50 clients.

| Acc(%) | Algorithm | CIFAR-100 | Tiny-Imagenet | Flower102 |
|---|---|---|---|---|
| | FedAvg | 17.57 | 14.38 | 22.81 |
| | Per-FedAvg | 28.08 | 25.49 | 41.52 |
| HmFL | FedProx | 18.11 | 14.18 | 21.66 |
| | FedPer | 30.51 | 26.42 | 47.86 |
| | **FedFuse** | **35.33 (↑15.7%)** | **32.11 (↑21.5%)** | **48.95 (↑2.3%)** |
| | FedKD | 30.37 | 18.58 | 26.71 |
| | FedProto | 25.61 | 15.46 | 12.74 |
| HeteroFL | FedMRL | 30.91 | 16.12 | 26.32 |
| | FedTGP | 25.93 | 25.13 | 27.87 |
| | **FedFuse** | **36.53 (↑18.2%)** | **32.68 (↑26.8%)** | **39.48 (↑41.6%)** |

## E.2 IMPLEMENTATION DETAILS

All experiments were conducted using PyTorch version 2.1.0 on NVIDIA 4090D GPUs with CUDA 12.1. Key hyperparameters included:

- Optimizer: Adam for both client and server updates.
- Client Local Training Learning Rate ($\eta$): 0.001
- Server Aggregation Learning Rate ($\eta_g$): 0.001
- Client Personalization Update Learning Rate ($\eta_p$): 0.001
- Local Epochs ($E_c$): 1
- Server Epochs ($E_s$): 1
- Batch Size (Local Training): 64
- Batch Size (Server Aggregation / Local Update on $\mathcal{D}^b$): 64
- Temperature ($\tau$): 2.0
- MoE Experts ($E$): 100 for K=100, 50 for K=50, 10 for K=10
- MoE top-$k$ ($k$): 40 for K=100, 20 for K=50, 4 for K=10
- Communication Rounds ($T$): 100
- Client Participation Rate ($p$): 0.1 for K=100, 0.2 for K=50, 1.0 for K=10
- Public Dataset ($\mathcal{D}^b$): A subset of the corresponding dataset
- Data Heterogeneity ($\alpha$ for Dirichlet): As specified in Table 2 (0.05, 0.1, 0.5). For IID experiments, data was shuffled and distributed uniformly.

Baseline implementations were based on publicly available codebases where possible, adapted to the HeteroFL setting.

## E.3 ADDITIONAL RESULTS FOR ACCURACY COMPARISON

Tables 9 and 10 present the detailed accuracy comparisons on the three datasets for scenarios with 50 and 100 clients, respectively, complementing Table 1 in the main text. Figure 6 illustrates the training convergence curves under the model-homogeneous setting, serving as a baseline comparison for the heterogeneous results shown in Figure 5. As a supplement to the experimental results, Table 11 provides detailed outcomes on NLP datasets.

Table 10: Comparison with the SOTA methods on three datasets in 100 clients.

| Acc(%) | Algorithm | CIFAR-100 | Tiny-Imagenet | Flower102 |
|---|---|---|---|---|
| HmFL | FedAvg | 11.30 | 10.44 | 14.15 |
| | Per-FedAvg | 20.38 | 20.18 | 35.14 |
| | FedProx | 11.23 | 13.10 | 14.53 |
| | FedPer | 28.40 | 23.15 | 38.05 |
| | **FedFuse** | **29.80** (↑4.9%) | **33.44** (↑44.4%) | **41.95** (↑10.2%) |
| HeteroFL | FedKD | 16.75 | 13.28 | 18.42 |
| | FedProto | 18.75 | 12.04 | 19.90 |
| | FedMRL | 23.97 | 11.39 | 26.87 |
| | FedTGP | 16.70 | 26.88 | 21.78 |
| | **FedFuse** | **31.90** (↑33.1%) | **30.98** (↑22.5%) | **30.56** (↑13.7%) |

Table 11: Comparison with the SOTA methods on AGNews in 50 clients.

| Algorithm | FedKD | FedProto | FedMRL | FedTGP | Ours |
|---|---|---|---|---|---|
| Acc(%) | 94.80 | 77.68 | 94.82 | 94.76 | 96.11(↑23.7%) |

### E.4 ADDITIONAL RESULTS FOR ABLATION STUDIES

The absolute accuracy improvement from MoE may appear marginal, but the primary contribution of MoE resides in dynamic fusion and selective integration of heterogeneous knowledge, rather than direct enhancement of global accuracy (Table 12). Its advantages manifest in: 1) Stability under high heterogeneity: In extreme non-IID settings ($\alpha = 0.1$), MoE reduces inter-client variance by 40.36% compared to average aggregation (w/o MoE Aggregation). 2) Accuracy improvement for low-frequency clients: MoE enhances accuracy by 13.59% for clients with infrequent aggregation participation.

Table 12: Performance Comparison.

| Model | Client Accuracy Variance | Low-frequency Client Accuracy(%) |
|---|---|---|
| FedFuse | 18.03 (↑40.36) | 29.49 (↑13.59) |
| w/o MoE | 30.23 | 25.96 |

### E.5 ADDITIONAL RESULTS FOR RESOURCE OVERHEAD

We have conducted additional quantitative analysis comparing our method (Ours) with its ablated version that removes the MoE component (denoted as w/o MoE). The FLOPs incurred by the MoE module are modest, largely due to our use of a top-k routing mechanism that activates only a subset of experts per forward pass (Table 13). Memory consumption remains well within feasible limits, even with increasing model size or expert count(Table 14).

### E.6 SENSITIVITY EXPERIMENTS

We conducted a sensitivity analysis on the top-k hyperparameters, with detailed visualizations provided in Figure 3.

We set the total number of experts to match the number of clients, and set the top-k is equal to the 40% of experts number, which offers the maximum personalization capacity—each client can, in theory, have a dedicated expert. However, to ensure computational and memory efficiency, our gating network adopts a top-k sparse routing strategy, where only a subset of experts is activated for each client in each round. As illustrated in Figure 3, activating 40% of the total experts per round (i.e., k = 0.4 × total experts) achieves the best trade-off between performance and efficiency.

We have also conduct an additional sensitivity analysis on the distillation temperature parameter $\tau$ used in the reverse KL divergence loss (Table 15). Specifically, we vary $\tau$ in a reasonable range e.g.,0.5,1.0,2.0,5.0 and measure the resulting performance across multiple client models.

Table 13: Memory Comparison.

| Methods | FLOPs | Memory |
|---------|-------|--------|
| FedFuse | 100G | 808.08MB |
| w/o MoE | 99.1G | 726.41MB |

Table 14: Memory-client Numbers Results.

| Client Numbers | 10 | 20 | 50 | 100 | 200 | 500 |
|----------------|------|--------|--------|--------|--------|---------|
| Memory(MB) | 680.2 | 707.23 | 808.08 | 864.95 | 902.43 | 1958.63 |

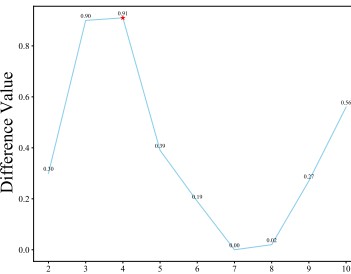

Figure 3: Difference Accuracy on different number of selected experts

Table 15: Time Complexity Analysis

| Temperature | 0.5 | 1 | 2 | 3 | 4 | 5 |
|-------------|-------|-------|-------|-------|-------|-------|
| **Accuracy(%)** | 33.56 | 40.35 | 42.22 | 39.22 | 38.84 | 34.75 |

The performance of our method remains stable over a broad range of $\tau$ values, indicating robustness to calibration variations. When $\tau$ is set to a moderate value (e.g., $\tau = 2.0$), the global logits are sufficiently softened, reducing the impact of overconfidence and improving the alignment with personalized local distributions. Lower $\tau$ values tend to retain sharper (and potentially overconfident) global predictions, slightly degrading performance, while excessively high $\tau$ values dilute the knowledge transfer.

These findings suggest that our reverse KL-based distillation mechanism is not overly sensitive to global model calibration, and that appropriate temperature tuning (e.g., $\tau \in [1.0, 2.0]$) can mitigate overconfidence issues in global logits.

## E.7 SCALABILITY WITH CLIENT NUMBERS.

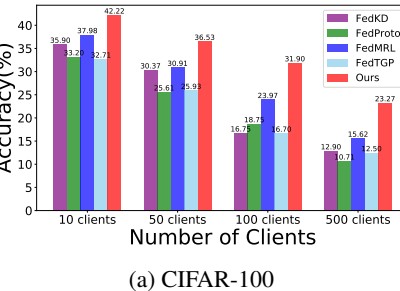
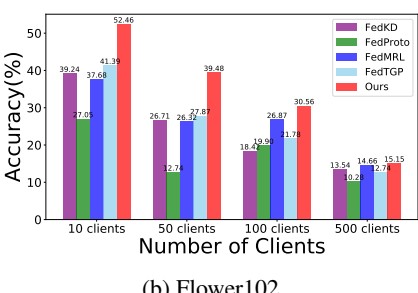

(a) CIFAR-100          (b) Flower102

Figure 4: Scalability: Accuracy vs. Number of Clients (K) on CIFAR-100 and Flower102 (HeteroFL Setting).

Figure 4 illustrates how the final average accuracy of different HeteroFL algorithms scales as the number of clients increases from $K = 10$ to $K = 500$. FedFuse exhibits greater robustness compared to baselines. For instance, on CIFAR-100, while other algorithms show a decline in accuracy with

more clients, FedFuse exhibits a more gradual decrease. On Flower102, FedFuse consistently achieves the highest accuracy across all client numbers. This highlights the robustness and scalability of FedFuse in federated learning scenarios.

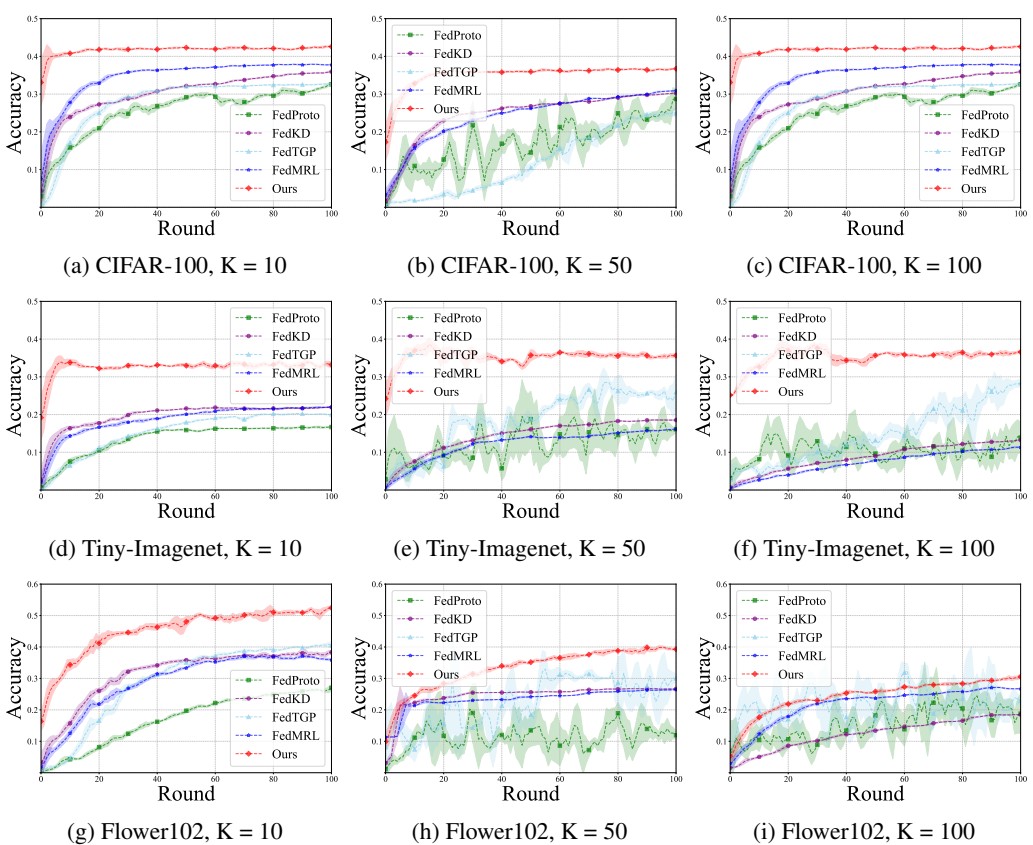

Figure 5: Training accuracy curves for various algorithms in heterogeneous federated learning (HeteroFL) across datasets and client numbers K.

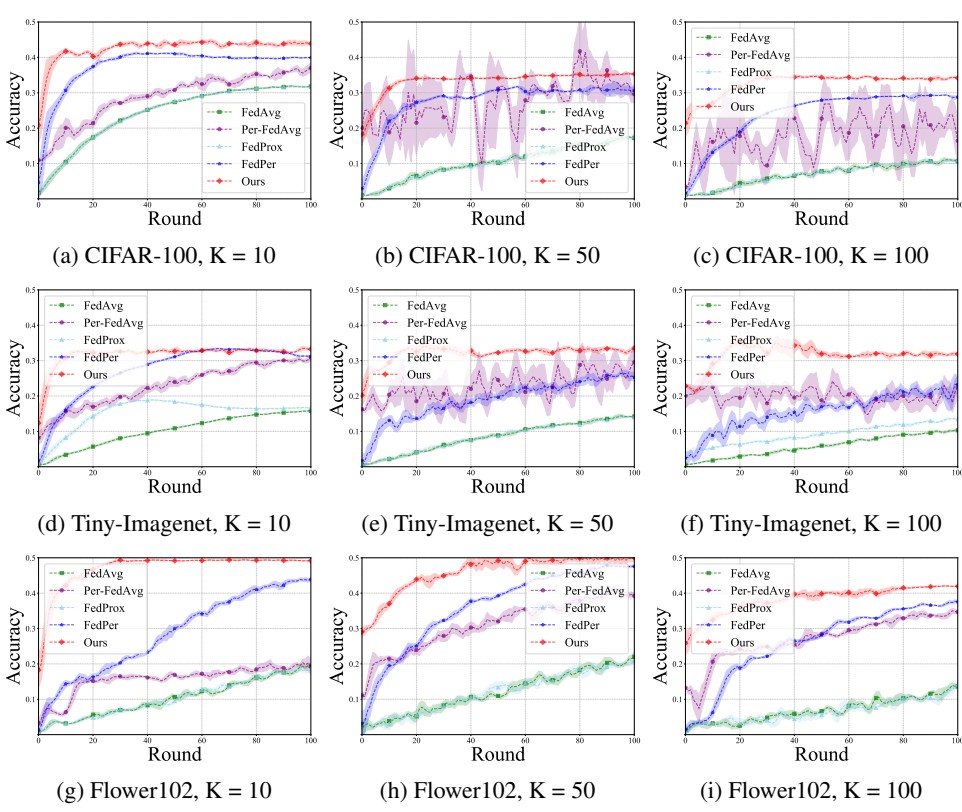

(a) CIFAR-100, K = 10      (b) CIFAR-100, K = 50      (c) CIFAR-100, K = 100

(d) Tiny-Imagenet, K = 10      (e) Tiny-Imagenet, K = 50      (f) Tiny-Imagenet, K = 100

(g) Flower102, K = 10      (h) Flower102, K = 50      (i) Flower102, K = 100

Figure 6: Training accuracy curves for various algorithms under the *homogeneous* federated learning setting (HmFL) across different datasets and client numbers (K). These serve as a baseline for evaluating personalization methods when architectural heterogeneity is absent.

