# OpenReview forum: "FedFuse: Selective Knowledge Distillation with Expert-Guided Fusion for Heterogeneous Federated"
_ICLR.cc/2026/Conference — ICLR 2026 Conference Withdrawn Submission_

### Official Review · Reviewer_J4d9 · 2025-10-17

**Soundness:** 2
**Presentation:** 3
**Contribution:** 2
**Rating:** 2
**Confidence:** 4

**Summary:**

The paper proposes a novel method for heterogeneous federated learning (FL), a setting in which FL clients can locally adopt model architectures of their choice. The proposed methodology relies on public data to transfer knowledge between clients and the server via logits computed locally on this shared transfer set. The main novelty compared to related work in the field lies in the use of a mixture-of-experts framework for server-side knowledge distillation.

**Strengths:**

Strength Points of the Paper

1. Novelty. The expert-guided fusion using a server-side MoE is a neat way to handle heterogeneity: it adaptively weights clients’ logits rather than averaging them.

2. Scale of the experimental settings. The paper provides experimental results with up to 500 clients.

3. Broad set of model architecture in heterogeneous FL setting for experiments.  The paper provides experimental results that consider a considerable set of model architecture in the federation.

**Weaknesses:**

Weak Points of the Paper

The main weaknesses of the paper can be summarized as:
(i) reliance on public data;

(ii) unclear contribution of the MoE aggregation to the empirical results (with MoE aggregation being the main novelty of the approach);

(iii) lack of clarity in baseline selection and ablation design.

In the following, I provide a more detailed discussion of these weaknesses.

The proposed mechanism relies on server-side knowledge distillation. To perform distillation, the method requires access to public data. This aspect is overlooked in the paper, despite being fundamental to its design and overall applicability.
The following questions and concerns arise:

1. Nature of the public data. In the experiments (Appendix E.2), the public data are described as “a subset of the corresponding dataset.” This seems a strong assumption in a federated setting, where data are private. It is unlikely to have public data that exactly match the semantic, characteristic, and distributional properties of the overall dataset (i.e., the union of all clients’ datasets). Furthermore, previous works have discussed the effectiveness of arbitrary transfer set (e.g., [Nayak et al.]), demonstrating that the transfer set should be "class balanced" for the considered task.

2. Size of the public dataset.
There is no discussion of the size of the public dataset (the authors only mention it as “small”).

3. Strategic advantage from public data.
Assuming the existence of a public dataset provides a strategic advantage compared to other methods that do not make this assumption (e.g., standard FedAvg).
For instance, prior work has demonstrated that in heterogeneous data settings, sharing even a small portion of data among clients can improve overall performance [Zhao et al.].

4. Privacy implications of shared logits
Clients send logits over public data, which could still reveal private information (e.g., via membership inference or model inversion attacks).
Since participating clients know the shared public data on which these logits are computed, this issue deserves at least a brief discussion in the paper.

5. Late introduction of the public-data assumption.
The requirement of public data (a key design choice) emerges only in the method section.
This assumption should be clearly stated and justified already in the abstract and introduction.

6. Realism of the public-data assumption.
The authors themselves acknowledge that “While this assumption [access to a small, publicly available dataset] may limit applicability in scenarios where no suitable public data exist, it is commonly adopted in federated learning literature and reflects practical scenarios.”
However, this assumption may still be unrealistic in many privacy-sensitive or domain-specific settings (e.g., healthcare, industrial FL), where no representative public data exist.

Other concerns are related to the following:

7. Lack of uncertainty reporting: Standard deviations are missing from key numerical results (Tables 1–3), limiting the assessment of statistical reliability.

8. Unclear contribution of MoE aggregation: The performance gain attributable to the MoE component is not clearly demonstrated (Table 3).

9. Baseline selection and ablation clarity: The criteria for choosing baselines from the related work section are not explained, and the setup for the “w/o Personalized Update” ablation (Section 4.2) is insufficiently detailed.

References

[Nayak et al.] G. K. Nayak, K. R. Mopuri, and A. Chakraborty, “Effectiveness of Arbitrary Transfer Sets for Data-Free Knowledge Distillation,” Proceedings of the IEEE/CVF Winter Conference on Applications of Computer Vision, 2021, pp. 1430–1438.

[Zhao et al.] Y. Zhao, M. Li, L. Lai, N. Suda, D. Civin, and V. Chandra, “Federated Learning with Non-IID Data,” arXiv preprint arXiv:1806.00582, 2018.

**Questions:**

For the rebuttal, I am looking forward to the following aspects and clarifications:

1. How is the public dataset produced? Is it independently and identically distributed? Is it class-balanced? Please provide details on how the public dataset is obtained or constructed. Consider to add this information in the main text and not only in Appendix.

2. How does the size of the public dataset influence performance?

3. How would potential skew (either semantic or in class distribution) affect performance?

4. Please justify or discuss the strategic advantage derived from using public data. I find the following questions particularly interesting (though I am not explicitly expecting these experiments if other justifications can be provided for the same point):
(a) How would FedAvg + shared data perform when using the same shared data as in FedFuse?
(b) What if all the clients perform pre-training on the public data (one time before the FL training starts)?

5. Why is the requirement for public data introduced only in the method section rather than discussed in the abstract and introduction? Please consider presenting this design choice earlier in the paper.

6. How realistic is the assumption of access to a small, publicly available dataset in privacy-sensitive or domain-specific scenarios?

7. What are the privacy implications of clients sharing logits over public data? Could this enable membership inference or model inversion attacks?

8. Consider to add standard deviations in the numerical results (Table 1, Table 2, Table 3). In particular, in Table 3 the entry "w/o MoE Aggregation" has performance close to the full variant of FedFuse. Here, uncertainty interval is foundamental to understand whether MoE Aggregation is relevant.

9. Since the results from Table 3 for "FedFuse (Full)" and "w/o MoE Aggregation" are close (at least with respect to the other entries), consider to extend this ablation to other settings to confirm the relevance of habing MoE Aggregation. Given the fact that MoE aggregation seems to be the main novelty introduced by the paper, clarifying this point is of paramount importance.

10. The authors state that “For statistical heterogeneity evaluation, we use the Dirichlet distribution with parameter $\alpha$ to partition data among clients.” However, typically, a lower $\alpha$ value corresponds to higher data heterogeneity. This does not appear to be the case in the reported results, where (e.g., Table 2) all baselines show generally better performance for lower $\alpha$ values. Please consider being more precise in describing how data is partitioned across clients following the Dirichlet allocation.

11. It is not clear how the baselines used in experiments have been selected among the works reported in Related Work. In particular, why the work in [Sattler et al.], already referenced in the paper's related work, has not been considered for comparison?

12. With reference to Section 4.2, it is not clear how the method operates in the case of “w/o Personalized Update”. In this setting, how do the clients collaboratively learn? Do they train only on their local data, independently?

13. Consider to improve attached code clarity (better instructions) to ensure direct reproducibility of results.

14. Authors state (in method section) that "The choice of reverse KL divergence is intentional and crucial for personalization". While this seems reasonable, I would encourage to also demonstrate this empirically (that reverse KL is better than regular one).

Minor aspects and suggestions:
* A description of how the considered baselines work can be useful.

* Provide the hyper-parameter tuning of baselines used in experiments. In particular, since hyper-parameters like the temperature (also needed in other considered baselines) are relevant for the results (as highlighted in Table 15), ensure that the comparison with other methods that use temperature is fair in selecting the best temperature tuning.

* Fig. 3 in Appendix has unclear x-axis label (which I think should represent what subset of experts are activated). Currently, it can be misleading.

* The caption of Table 15 ("Time Complexity Analysis") is wrong (should be about temperature tuning).


References

[Zhao et al.] Y. Zhao, M. Li, L. Lai, N. Suda, D. Civin, and V. Chandra, “Federated Learning with Non-IID Data,” arXiv preprint arXiv:1806.00582, 2018.

[Sattler et al.] Felix Sattler, Arturo Marban, Roman Rischke, and Wojciech Samek. CFD: Communication-efficient
federated distillation via soft-label quantization and delta coding. IEEE Transactions on Network
Science and Engineering, 9(4):2025–2038, 2021

---

### Official Review · Reviewer_Sy1U · 2025-10-23

**Soundness:** 2
**Presentation:** 3
**Contribution:** 2
**Rating:** 2
**Confidence:** 4

**Summary:**

The paper proposes FedFuse, a federated learning (FL) framework that addresses personalization and model heterogeneity across clients. The central idea is to combine two mechanisms: (1) an Expert-Guided Fusion (EGF) strategy based on a Mixture-of-Experts (MoE) module at the server to adaptively aggregate client knowledge, and (2) a Selective Knowledge Distillation (SKD) method that allows clients to selectively integrate global knowledge relevant to their local data distributions. Instead of sharing model parameters, clients transmit logits computed from a small public dataset (Db), which the server uses to perform adaptive knowledge fusion.

Experiments are conducted across four datasets (CIFAR-100, Tiny-ImageNet, Flower102, and AGNews) with different numbers of clients (10, 50, 100, and 500) and compared against several baseline methods, including FedKD, FedProto, FedMRL, and FedTGP. The authors report significant improvements in accuracy and reduced communication costs relative to these baselines. Theoretical convergence results are presented for convex and non-convex cases, supported by standard assumptions.

The topic is timely, and the paper is clearly written with a good logical flow. The combination of MoE-based server aggregation and selective distillation is an intuitive extension of existing approaches and could provide practical benefits in settings with heterogeneous client architectures. However, the contribution, while interesting, lacks the methodological and experimental rigor expected of a top-tier conference submission.

**Strengths:**

The paper addresses an important problem in federated learning, i.e., balancing personalization and efficiency under model heterogeneity.
•	The combination of Expert-Guided Fusion and Selective Knowledge Distillation is conceptually interesting and intuitively motivated.
•	The experiments include both computer vision and NLP datasets, providing some cross-domain evidence of applicability.
•	The manuscript is generally well-structured and easy to follow, with clear motivation and logical organization.

**Weaknesses:**

Reliance on a shared public dataset (Db): The proposed method depends on the existence of a public dataset accessible to both clients and the server for generating logits. This assumption is unrealistic in many federated learning scenarios, where privacy restrictions or domain isolation make shared data unavailable. Even with availability of such a public dataset, whether it is a good representation of all local datasets is an important concern and can potentially disrupt the performance of KD process by inducing bias in the trained models. Thus, the method’s applicability in practice is limited, and the reported results may not generalize to realistic settings.

2) Scalability limitations due to the number of experts (E = K): As described in Appendix E.2, the number of experts in the server-side MOE module equals the number of clients (E = K). This configuration causes the server’s computational and memory cost to scale linearly with the number of participants, rendering the approach unscalable for large federated networks. Assigning one expert per client also undermines the idea of shared expert specialization and turns the MoE into a client-indexed ensemble rather than a true expert system.

3) Reproducibility concerns regarding the public dataset: The paper does not specify how the public dataset (Db) is selected and whether it is from the original public or the distributed data from clients and with what portion. Since the performance of distillation-based federated learning methods is highly sensitive to the characteristics of the public data, explicit dataset selection criteria and preprocessing details are necessary for reproducibility.

4) Presentation and writing inconsistencies The paper contains several grammatical and stylistic inconsistencies. For example, the abbreviation MoE is used before being spelled out, while later in the text the term Mixture-of-Experts (MoE) is redundantly written in full multiple times. Such inconsistencies suggest limited editorial care and may confuse readers unfamiliar with the terminology. Careful proofreading is needed to correct these minor issues and improve overall presentation quality.

**Questions:**

Although there are a number of possible ways to improve the manuscript, my recommendation to the authors is to focus, with priority in addressing the limitation outlined in (3) above. The lack of reproducibility renders any contribution unusable.

---

### Official Review · Reviewer_hAF5 · 2025-11-01

**Soundness:** 3
**Presentation:** 3
**Contribution:** 2
**Rating:** 4
**Confidence:** 2

**Summary:**

This paper proposes FedFuse to address the loss of personalized knowledge and model divergence during knowledge fusion and aggregation. Specifically, FedFuse combines: (1) a server-side expert-guided fusion built on a mixture-of-experts (MoE) model to adaptively aggregate heterogeneous client knowledge in the logits space, and (2) a client-side selective knowledge distillation process that uses reverse KL divergence to update clients from adaptive global logits while preserving local model specificity.
By convergence proofs over both convex and non-convex cases, and experiments over standard benchmarks, including CIFAR-100, TinyImageNet, Flower102, AGNews, this paper demonstrates its effectiveness.

**Strengths:**

1. This paper is well-organized, both the algorithm, figure clearly illustrates this work, which employ of an MoE-based fusion at the server for logits aggregation, and a reverse KL divergence-based personalization on the client for adaptive optimization.
2. Experimental results are significant, the experiments span diverce datasets with varying client numbers, and heterogeneity levels. The exeprients in Table 1 and 2 clearly show large accuracy improvements in high heterogeneity regimes.
3. Convergence proofs are provided and appear logically consistent, with assumptions, lemmas, and bounds shown both for the convex and non-convex scenarios. These proofs are clear.

**Weaknesses:**

1. Limited discussion on MoE stability and gating strategy: Although Figure 3 and Section E.6 analyze the impact of varying the top-k experts in the MoE, there is little discussion on how stable the routing/gating mechanism is over time. Specifically, whether certain clients or features consistently dominate some experts, or whether the gating network can overfit to client idiosyncrasies. This has practical consequences for the ability of the MoE to generalize beyond seen public samples, especially in highly imbalanced data settings.
2. The evaluation relies on suitable mini-size public datasets. However, the data size and evaluated domains are limited, the author could include more data size or domains for comprehensive evaluation, e.g. diverse domains by following CoOp[1].
3. This paper could report more detailed comparisons, except for the accuracy/memory/time scaling, such as the network failures, delayed updates, especially for larger clients.

**Questions:**

My question is for your MoE designs and analysis, specifically:
1. Expert gating analysis: Could the authors provide an analysis of per-expert activation frequencies and the stability of the gating network across rounds/clients? For example, do specific experts become overspecialized or underutilized?
2. MoE design choices: What is the impact of the number of experts VS the number of clients? The paper chooses the number of experts close to or equal to the client count, however, in real-time scenarios, the number of client should be larger than server, would significant over/under number of experts meaningfully harm fusion or introduce bottlenecks?

---

### Official Review · Reviewer_JGik · 2025-11-02

**Soundness:** 2
**Presentation:** 3
**Contribution:** 2
**Rating:** 4
**Confidence:** 3

**Summary:**

The paper presents a well-motivated approach to a challenging problem in Federated Learning, introducing a novel combination of MoE-based aggregation and reverse KL divergence for personalization. The strengths of the work lie in its methodological novelty, supported by extensive empirical validation across multiple datasets, client scales, and heterogeneity levels, as well as a theoretical analysis. The ablation studies effectively justify the core components. However, the method's reliance on a public dataset for knowledge representation is a notable limitation, and its applicability in scenarios without such data is not thoroughly discussed. Furthermore, while scalability is tested via simulation, evidence of practical deployment on physical devices or with extremely large models is absent, as acknowledged by the authors.

**Strengths:**

The paper provides extensive empirical evidence across multiple benchmarks, varying numbers of clients, and different levels of statistical heterogeneity, demonstrating consistent and often significant performance gains over a wide range of state-of-the-art baselines.

FedFuse shows superior performance and stability under conditions of high data heterogeneity (low α values), with its advantage often increasing as heterogeneity becomes more severe.

The ablation experiments clearly demonstrate the contribution of each key component, validating the design choices of the proposed framework.

The provision of convergence guarantees for both strongly convex and non-convex cases under standard federated optimization assumptions adds theoretical rigor and supports the stability of the proposed algorithm

**Weaknesses:**

The method's core mechanism relies on the existence of a small, public dataset ($D^b$) for computing and fusing logits. The paper does not explore the sensitivity of performance to the choice, size, or domain relevance of this public dataset, nor does it discuss fallback strategies or the method's applicability in domains where such data is unavailable.

Although a wide range of baselines is compared, the analysis could be deepened by providing more explicit reasoning for the performance gaps, especially for the weaker baselines. For instance, why FedProto struggles significantly on some datasets (e.g., Flower102 with 50 clients in Table 9) is not discussed in detail.

The paper demonstrates that the MoE mechanism improves performance, but it does not provide direct analysis or visualization (e.g., of expert weights or routing patterns) to illustrate how different experts specialize in knowledge from different types of clients or data distributions, which would offer deeper insight into the fusion process.

While a sensitivity analysis on the top-k gating parameter is provided in the apendix, the justification for the number of experts (E) used in different scenarios (e.g., E=100 for K=100) is less clear, and the impact of these choices on performance and resource overhead could be further elaborated.

**Questions:**

See Weaknesses

---

### Official Review · Reviewer_KxHE · 2025-11-04

**Soundness:** 1
**Presentation:** 1
**Contribution:** 1
**Rating:** 2
**Confidence:** 4

**Summary:**

The paper proposes “FedFuse,” a federated learning approach where clients train locally, upload public-data logits to the server, the server trains a mixture-of-experts (MoE) with a gating network to fuse these client logits, and then returns “refined” global logits. Clients finally perform a “selective” personalization step by distilling from the server’s logits. The method targets both statistical and architectural heterogeneity and is evaluated on image and text datasets with varying numbers of clients.

**Strengths:**

- Reasonable problem framing: fusing heterogeneous client knowledge while preserving personalization is important in practice.
- Server-side MoE over logits is a sensible way to adapt to heterogeneous client outputs without sharing weights or raw data.
- Communication format (logits on public data) is practical; experiments cover several datasets and client counts.
- The pipeline is modular and compatible with diverse local models.

**Weaknesses:**

- Novelty is limited. The overall recipe—server-side fusion on public data via distillation plus client-side personalization—closely resembles prior work. In particular, FedDF (Lin et al., NeurIPS 2020) already studies ensemble/distillation-based server fusion on public data; FedFed (Yang et al., NeurIPS 2023) addresses representation/feature distillation under heterogeneity; and TAKFL (Morafah et al., NeurIPS 2024) directly tackles selective knowledge integration and information dilution under device/model heterogeneity. The paper cites some of these but does not meaningfully compare or position against them, and many of the claimed insights (selective transfer, preservation of personalized features, handling architecture/data heterogeneity) overlap with those works.
- Naming confusion. Using “HeteroFL” as a label for your setting is easily confused with the specific method “HeteroFL” (Diao et al., ICLR 2021). Please use “model-heterogeneous FL (HFL)” for the setting and reserve “HeteroFL” for the cited method.
- Related work is thin and dated. The related work section misses recent methods on lightweight representations, distillation, subspace/model transformation, and the FedMoE literature. This makes the framing feel incomplete.
- Notation and missing definitions. The superscript “r” in D_i^r is never defined when first introduced. Equation (2) refers to L_P but the paper doesn’t clearly define it at that point (it reads like a standard cross-entropy or personalization loss). These issues slow understanding.
- “Selective” KD is not actually selective. The client step appears to be a straight reverse-KL distillation from server logits; there is no explicit selection criterion (e.g., instance/class gating, confidence/consistency filtering, feature alignment). As written, the selectivity claim is not supported by the method.
- Server-side training/gating is under-specified. It’s unclear exactly how the gating network is optimized, what the objective encourages (e.g., specialization vs averaging), whether any load-balancing/entropy regularization is used, and whether the server architecture differs from client models. The link to the FedMoE literature isn’t made, so it’s hard to see what’s new in the routing/fusion design.
- Complexity and ablations. The pipeline adds multiple moving parts (server-side MoE/gating, client reverse-KL personalization, public-data orchestration), but there’s no clear compute/communication analysis or end-to-end scaling study. Ablations isolating where the gains come from (MoE, gating/top-k, temperatures, client personalization) are missing, so the source of improvements is unclear.
- Convergence section adds standard assumptions/bounds and does not yield insight specific to this MoE-plus-distillation setup; it reads as boilerplate.
- Ambiguous target (personalized vs global). At points the text reads like a global fusion objective; elsewhere it’s personalized FL. Please crisply state the target (per-client personalized performance) and report metrics accordingly (mean/variance across clients).
- Experimental design and reporting details are insufficient. The exact public dataset(s), local epochs, client sample rate, and budget parity across baselines are not clearly documented. Using local epoch e=1 with ~100 rounds is atypical for KD-based methods; prior work (e.g., FedDF) often needs more local compute to produce stable, informative logits. With e=1 and few rounds, strong distillation baselines are likely underpowered, which casts doubt on the fairness of the comparisons. Also, participation rate should be chosen with the number of clients in mind; otherwise you inadvertently reduce heterogeneity and make the task easier. See Morafah et al., IEEE TAI 2023, for principled FL experimental design.
- Terminology/labels. “HmFL” appears without definition; “HeteroFL” is used ambiguously (method vs setting). These small issues accumulate and hurt clarity.

Overall, the method and presentation feel loose: several important design choices are not justified, the “selective” claim is unsupported by the algorithm, and the comparison to the most relevant recent work is missing.

**Questions:**

1. What, concretely, is the selection mechanism in “selective KD”? If selection is implicit (just reverse-KL), consider either adding an explicit selection criterion or revising the claim.

2. How exactly is the server gating trained (objective, regularizers, load-balancing, top-k vs soft routing)? Does the server model differ architecturally from clients?

3. What public data is used and how is it constructed? Any class-coverage or distribution overlap issues with client test sets? Please report sensitivity to public-set size and distribution shift.

4. Can you provide a strict compute/communication complexity analysis and scaling curves vs number of clients, public-set size, number of experts, and routing sparsity (k)?

5. Please clarify the target: is the goal personalized FL? If so, ensure metrics reflect per-client performance and dispersion.

6. Why choose e=1 local epoch and ~100 rounds across all methods? Re-run key comparisons with larger e and more rounds (as common for KD methods) and report multiple seeds with confidence intervals.

7. How does the approach relate empirically and conceptually to TAKFL (task-arithmetic knowledge integration), FedDF (server-side ensemble distillation), and FedFed (feature/representation distillation)? These are close baselines in spirit and should be included with careful tuning parity.

---

### Note · Authors · 2025-11-24

**Comment:**

All authors agree to withdraw this manuscript.

**Withdrawal Confirmation:**

I have read and agree with the venue's withdrawal policy on behalf of myself and my co-authors.